# High-dimensional Asymptotics of VAEs: Threshold of Posterior Collapse and Dataset-Size Dependence of Rate-Distortion Curve

## Abstract

In variational autoencoders (VAEs), the variational posterior often aligns closely with the prior, known as posterior collapse, which leads to poor representation learning quality. An adjustable hyperparameter beta has been introduced in VAE to address this issue. This study sharply evaluates the conditions under which the posterior collapse occurs with respect to beta and dataset size by analyzing a minimal VAE in a high-dimensional limit. Additionally, this setting enables the evaluation of the rate-distortion curve in the VAE. This result shows that, unlike typical regularization parameters, VAEs face "inevitable posterior collapse" beyond a certain beta threshold, regardless of dataset size. The dataset-size dependence of the derived rate-distortion curve also suggests that relatively large datasets are required to achieve a rate-distortion curve with high rates. These results robustly explain generalization behavior across various real datasets with highly non-linear VAEs.

## 1 Introduction

Deep latent variable models are generative models that use a neural network to convert latent variables generated from a prior distribution into samples that closely resemble the data. Variational autoencoders (VAEs) (Kingma & Welling, 2013; Rezende et al., 2014), a type of the deep latent variable models, have been applied in various fields such as image generation (Child, 2020; Vahdat & Kautz, 2020), clustering (Jiang et al., 2016), dimensionality reduction (Akkari et al., 2022), and anomaly detection (An & Cho, 2015; Park et al., 2022). In VAEs, directly maximizing the likelihood is intractable owing to the marginalization of latent variables. Therefore, VAE often employs the evidence lower bounds (ELBOs), which serve as computable lower bounds for the log-likelihood.

From an informational-theoretical perspective, several studies (Alemi et al., 2018; Huang et al., 2020; Nakagawa et al., 2021) have interpreted ELBO as decomposing into two terms that represent a trade-off. Based on the analogy from the rate-distortion (RD) theory, these terms can be likened to *rate* and *distortion* (Alemi et al., 2018). Furthermore, these studies suggest that during training with ELBO, the variational posterior of the latent variables tends to align with their prior, hindering effective representation learning. This phenomenon is commonly referred to as "posterior collapse".

To address the posterior collapse, an additional regularization parameter, denoted as $\beta_{\mathrm{VAE}}$, is introduced to control the trade-off between rate and distortion (Higgins et al., 2016). Although models with a small $\beta_{\mathrm{VAE}}$ can reconstruct the data points effectively, achieving low distortion, they may generate inauthentic data due to significant mismatches between the variational posterior and the prior (Alemi et al., 2018). In contrast, while models with a large $\beta_{\mathrm{VAE}}$ align their variational distributions closely with the prior, resulting in a low rate, they may ignore the important encoding information. Thus, careful tuning of $\beta_{\mathrm{VAE}}$ in beta-VAEs is important for various applications (Kohl et al., 2018; Castrejon et al., 2019). In addition to simply enhancing the data generation capability, $\beta_{\mathrm{VAE}}$ is crucial for achieving better disentanglement (Higgins et al., 2016) and obtaining the RD curve (Alemi et al., 2018). However, theoretical understanding of the relationship between $\beta_{\mathrm{VAE}}$, the posterior collapse, and the RD curve remains limited. Particularly, the dataset-size dependence of these matters remains theoretically unexplored, even for linear VAE (Lucas et al., 2019).

**Contributions** This study advances the theory regarding dataset and $\beta_{\text{VAE}}$ dependence of the conditions leading to the posterior collapse and the RD curve, using a minimal model, referred to as the linear VAE (Lucas et al., 2019), which captures the core behavior of beta-VAEs even for more complex deep models (Bae et al., 2022). Throughout the manuscript, this study considers a high-dimensional limit, where both the number of training data $n$ and dimension $d$ are large $(n, d \to \infty)$ while remaining comparable, i.e., $\alpha \triangleq n/d = \Theta(n^0)$. Our main contributions are:

- The dataset-size dependence of generalization properties, RD curve, and posterior-collapse metric in the VAE is sharply characterized by a small finite number of summary statistics, derived using high-dimensional asymptotic theory. Using these summary statistics, three distinct phases are characterized, pinpointing the boundary of the posterior collapse.

- A phenomenon where the generalization error peaks at a certain sample complexity $\alpha$ for a small $\beta_{\text{VAE}}$ is observed. As $\beta_{\text{VAE}}$ increases, the peak gradually diminishes, which is similar to the interpolation peak in supervised regression for the regularization parameter.

- Our analysis reveals "inevitable posterior collapse". A long plateau in the signal recovery error exists with respect to the sample complexity $\alpha$ for a large $\beta_{\text{VAE}}$. As $\beta_{\text{VAE}}$ increases, the plateau extends and eventually becomes infinite, regardless of the value of the sample complexity. These results are experimentally robust for real datasets with nonlinear VAEs.

- With an infinite dataset size limit, the RD curve, introduced from the analogy of the RD theory, is confirmed to coincide exactly with that of the Gaussian sources. Furthermore, the RD curve is evaluated for various sample complexities, revealing that a larger dataset is required to achieve an optimal RD curve in the high-rate and low-distortion regions.

The code used in this manuscript is submitted as supplemental material along with this manuscript.

**Notation** Here, we summarize the notations used in this study. The expression $\| \cdot \|_F$ denotes the Frobenius norm. The notation $\oplus$ denotes the concatenation of vectors; for vectors $\boldsymbol{a} \in \mathbb{R}^d$ and $\boldsymbol{b} \in \mathbb{R}^k$, $\boldsymbol{a} \oplus \boldsymbol{b} = (a_1, \ldots, a_d, b_1, \ldots, b_k)^\top \in \mathbb{R}^{d+k}$. $I_d \in \mathbb{R}^{d \times d}$ denotes an $d \times d$ identity matrix, and $\mathbf{1}_d$ denotes the vector $(1, \ldots, 1)^\top \in \mathbb{R}^d$ and $\mathbf{0}_d$ denotes the vector $(0, \ldots, 0)^\top \in \mathbb{R}^d$. $D_{\text{KL}}[\cdot\|\cdot]$ denotes the Kullback–Leibler (KL) divergence. For the matrix $A = (A_{ij}) \in \mathbb{R}^{d \times k}$ and a vector $\boldsymbol{a} = (a_i) \in \mathbb{R}^d$, we use the shorthand expressions $dA \triangleq \prod_{i=1}^{d} \prod_{j=1}^{k} dA_{ij}$ and $d\boldsymbol{a} \triangleq \prod_{i=1}^{d} da_i$, respectively. For vector $\boldsymbol{a} \in \mathbb{R}^d$, we also use the expression $\boldsymbol{a}_{:k} = (a_1, \ldots, a_k) \in \mathbb{R}^k$ where $k \leq d$.

## 2 RELATED WORK

**Linear VAEs** The linear VAE is a simple model in which both the encoder and decoder are constrained to be affine transformations (Lucas et al., 2019). Although deriving analytical results for deep latent models is intractable, linear VAEs can provide analytical results, facilitating a deeper understanding of VAEs. Indeed, despite their simplicity, the results in linear VAEs sufficiently explain the behavior of more complex VAEs (Lucas et al., 2019; Bae et al., 2022). Moreover, an algorithm proven effective for linear VAEs has been successfully applied to deeper models (Bae et al., 2022). In addition, various theoretical results have been obtained. Dai et al. (2018) demonstrated the connections between linear VAE, probabilistic principal component analysis (PCA) (Tipping & Bishop, 1999), and robust PCA (Candès et al., 2011; Chandrasekaran et al., 2011). Simultaneously, Lucas et al. (2019) and Wang & Ziyin (2022) employ linear VAEs to explore the origins of posterior collapse. However, these analyses did not address the dataset-size dependence of the generalization, RD curve, and robustness against the background noise, which is a focus of our study. Additionally, these analyses did not examine the behavior of the RD curve, which can be obtained by varying $\beta_{\text{VAE}}$ with a fixed decoder variance.

**High-dimensional asymptotics from replica method** The replica method, mainly used as an analytical tool in our study, is a non-rigorous but powerful heuristic in statistical physics (Mézard et al., 1987; Mezard & Montanari, 2009; Edwards & Anderson, 1975). This method has proven invaluable in solving high-dimensional machine-learning problems. Previous studies have addressed the dataset-size dependence of the generalization error in supervised learning including single-layer (Gardner & Derrida, 1988; Opper & Haussler, 1991; Barbier et al., 2019; Aubin et al., 2020) and

multi-layer (Aubin et al., 2018) neural networks, as well as kernel methods(Dietrich et al., 1999; Bordelon et al., 2020; Gerace et al., 2020). In unsupervised learning, this includes dimensionality reduction techniques such as the PCA (Biehl & Mietzner, 1993; Hoyle & Rattray, 2004; Ipsen & Hansen, 2019), and generative models such as energy-based models (Decelle et al., 2018; Ichikawa & Hukushima, 2022) and denoising autoencoders (Cui & Zdeborová, 2023). However, the dataset-size dependence of VAEs has yet to be previously analyzed; therefore, this study aims to examine this dependence. Efforts have been made to confirm the non-rigorous results of the replica method using other rigorous analytical techniques. For convex optimization problems, the Gaussian min-max theorem (Gordon, 1985; Mignacco et al., 2020) can be used in the analysis, which provides rigorous results consistent with those of the replica method (Thrampoulidis et al., 2018).

# 3 BACKGROUND

## 3.1 VARIATIONAL AUTOENCODERS

The VAE (Kingma & Welling, 2013) is a latent generative model. Let $\mathcal{D} = \{\boldsymbol{x}^\mu\}_{\mu=1}^n$ be the training data, where $\boldsymbol{x}^\mu \in \mathbb{R}^d$ and $p_\mathcal{D}(\boldsymbol{x})$ is the empirical distribution of the training dataset. In practical applications, VAEs are typically trained using beta-VAE objective (Higgins et al., 2016) given by

$$\mathbb{E}_{p_\mathcal{D}} \left[ \mathbb{E}_{q_\phi}[-\log p_\theta(\boldsymbol{x}|\boldsymbol{z})] + \beta_{\text{VAE}} D_{\text{KL}}[q_\phi(\boldsymbol{z}|\boldsymbol{x})\|p(\boldsymbol{z})] \right] \triangleq \mathbb{E}_{p_\mathcal{D}}[\mathcal{L}(\theta, \phi; \boldsymbol{x}, \beta_{\text{VAE}})], \quad (1)$$

where $\boldsymbol{z} \in \mathbb{R}^k$ is the latent variables and $p(\boldsymbol{z})$ is a prior for the variables, and the parameter $\beta_{\text{VAE}} \geq 0$ is introduced to control the trade-off between the first and second terms. $p_\theta(\boldsymbol{x}|\boldsymbol{z})$, parameterized by parameters $\theta$, and $q_\phi(\boldsymbol{z}|\boldsymbol{x})$, parameterized by $\phi$, are commonly referred to as *decoder* and *encoder*, respectively. Subsequently, VAEs optimize the encoder parameters $\phi$ and decoder parameters $\theta$ by minimizing the objective of Eq. (1). Note that when $\beta_{\text{VAE}} = 0$, the objective is a deterministic autoencoder that focuses only on minimizing the first term, which is referred to as the *reconstruction error*.

## 3.2 INFORMATION-THEORETIC INTERPRETATION OF VAEs

Alemi et al. (2018); Huang et al. (2020); Park et al. (2022) demonstrate that VAEs can be interpreted based on the RD theory (Davisson, 1972; Cover, 1999), which has been successfully applied to data compression. The primary focus has been on the curve where the distortion achieves its minimum value for a given rate, or conversely; see Appendix B for a detailed explanation. Based on an analogy from the RD theory, Alemi et al. (2018) decomposed the beta-VAE objective in Eq. (1) into *rate $R$* and *distortion $D$* as follows:

$$R(\phi) = \mathbb{E}_{p_\mathcal{D}}[D_{\text{KL}}[q_\phi(\boldsymbol{z}|\boldsymbol{x})\|p(\boldsymbol{z})]], \quad D(\theta, \phi) = \mathbb{E}_{p_\mathcal{D}}[\mathbb{E}_{q_\phi}[-\log p_\theta(\boldsymbol{x}|\boldsymbol{z})]]. \quad (2)$$

According to Alemi et al. (2018), a trade-off exists between the rate and distortion, as in the RD theory, especially when the encoder and decoder have infinite capacities. This relationship is derived from the following:

$$H = -\mathbb{E}_{p_\mathcal{D}} D_{\text{KL}}[q_\phi(\boldsymbol{z}|\boldsymbol{x})\|p_\theta(\boldsymbol{z}|\boldsymbol{x})] + R(\phi) + D(\theta, \phi),$$

where $H$ is the negative log-likelihood, defined as $H = -\mathbb{E}_{p_\mathcal{D}} \log p_\theta(\boldsymbol{x})$. From the non-negativity of the KL divergence, it follows that $H \leq R(\phi) + D(\theta, \phi)$, where the equality holds if and only if the variational posterior and true posterior coincide, i.e., $\forall \boldsymbol{x}, q_\phi(\boldsymbol{z}|\boldsymbol{x}) = p_\theta(\boldsymbol{z}|\boldsymbol{x})$.

While this equality holds when the encoder and decoder with infinite capabilities satisfy the optimality conditions, the limitation of finite parameters makes this situation unfeasible. Therefore, the goal is to determine an approximate optimal distortion at a given rate $R^*$ by solving the optimization problem:

$$\hat{D}(R^*) = \min_{\theta, \phi} D(\phi, \theta) \text{ s.t. } R(\phi) \leq R^*. \quad (3)$$

For optimization without explicitly considering $R^*$, the Lagrangian function with the Lagrange multiplier $\beta_{\text{VAE}} \geq 0$ can be utilized as follows:

$$\min_{\theta, \phi} D(\theta, \phi) + \beta_{\text{VAE}} R(\phi).$$

This formulation is identical to the beta-VAE objective expressed in Eq. (1). Thus, training various VAEs with different $\beta_{\text{VAE}}$ corresponds to obtaining distinct points on the RD curve.

# 4 SETTING

**Data model** We derive our theoretical results for dataset $\mathcal{D} = \{\boldsymbol{x}^\mu\}_{\mu=1}^n$ drawn from spiked covariance model (SCM) (Wishart, 1928; Potters & Bouchaud, 2020), which has been widely studied in statistics to analyze the performance of unsupervised learning methods such as PCA (Ipsen & Hansen, 2019; Biehl & Mietzner, 1993; Hoyle & Rattray, 2004), sparse PCA (Lesieur et al., 2015), and deterministic autoencoders (Refinetti & Goldt, 2022). The datasets are sampled according to

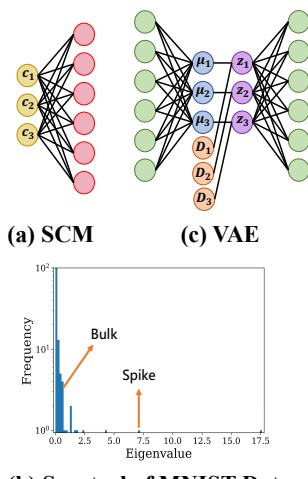

**(a) SCM**      **(c) VAE**

$$\boldsymbol{x}^\mu = \sqrt{\frac{\rho}{d}} W^* \boldsymbol{c}^\mu + \sqrt{\eta}\, \boldsymbol{n}^\mu, \ \ \forall \mu = 1, \dots, n, \quad (4)$$

where $W^* \in \mathbb{R}^{d \times k^*}$ is a deterministic unknown $k^*$ feature matrix, $\boldsymbol{c}^\mu \in \mathbb{R}^{k^*}$ is a random vector drawn from some distribution $p(\boldsymbol{c})$, $\boldsymbol{n}^\mu$ is a background noise vector whose components are i.i.d from the standard Gaussian distribution and $\rho \in \mathbb{R}$ and $\eta \in \mathbb{R}$ are scalar values to control the strength of the noise and signal, respectively. Different choices for $W^*$ and the distribution of $\boldsymbol{c}$ allow the modeling of Gaussian mixtures, sparse codes, and non-negative sparse coding. Note that, despite $W^*$ not being orthogonal, $W^* \boldsymbol{c}^\mu$ can be rewritten as $(W^* R)(R^{-1} \boldsymbol{c})$, where $R$ is a matrix that orthogonalizes and normalizes the columns of $W^*$. This can be considered as an equivalent system in which the new feature vector is $R^{-1} \boldsymbol{c}$. Therefore, without the loss of generality, we assume that $(W^*)^\top W^* = I_{k^*}$.

**(b) Spectral of MNIST Data**

Figure 1: The architectures of linear SCM (a) and VAE (c). The spectrum of the covariance matrix of the MNIST dataset (b) (Deng, 2012), which can be divided into a bulk and a finite number of spikes as in SCM.

**Spectrum of the covariance matrix of the dataset** The spectrum of the empirical covariance matrix of $\mathcal{D}$ is characterized by $W^*$ and $\boldsymbol{c}$. When $\boldsymbol{c}^\mu = 0$, the dataset are Gaussian vectors, whose empirical covariance matrix, with $n = \mathcal{O}(d)$ samples, follows a Marchenko-Pastur distribution characterized by the noise strength $\eta$ (Marchenko & Pastur, 1967). In contrast, by sampling $\boldsymbol{c} \sim p(\boldsymbol{c})$, the covariance matrix has $k^*$ eigenvalues, i.e., *spike*, with the columns of $W^*$ as the corresponding eigenvectors. The remaining $d - k^*$ eigenvalues, i.e., *bulk*, of the empirical covariance matrix still follow the Marchenko-Pastur distribution. This Spectrum is similar to that of the empirical covariance matrix of real datasets such as CIFAR10 (Krizhevsky et al.) and MNIST (Deng, 2012), as in Fig. 1 and further explained in Refinetti & Goldt (2022). Moreover, the validity of the assumption of SCM as a realistic data distribution has been supported by *Gaussian universality*, which indicates that the learning dynamics with real data, irrespective of the machine learning models, closely agree with those with the Gaussian model with the empirical covariance matrix of the data (Liao & Couillet, 2018; Mei & Montanari, 2022; Hu & Lu, 2022; Goldt et al., 2022).

**VAE model** In this study, we analyze the following two-layer VAE model:

$$p_W(\boldsymbol{x}|\boldsymbol{z}) = \mathcal{N}\left(\boldsymbol{x}; \frac{W\boldsymbol{z}}{\sqrt{d}}, \sigma^2 I_d\right), \ \ q_{V,D}(\boldsymbol{z}|\boldsymbol{x}) = \mathcal{N}\left(\boldsymbol{z}; \frac{V^\top \boldsymbol{x}}{\sqrt{d}}, D\right), \ \ p(\boldsymbol{z}) = \mathcal{N}(\boldsymbol{z}; \boldsymbol{0}_k, I_k). \quad (5)$$

The VAE in Eq. (5) is parameterized by the diagonal covariance matrix $D \in \mathbb{R}^{k \times k}$, and the weights $W \in \mathbb{R}^{d \times k}$ and $V \in \mathbb{R}^{d \times k}$, as shown in Fig. 1 (c). This model is called a linear VAE (Dai et al., 2018; Lucas et al., 2019; Sicks et al., 2021). In this study, we focus on the behavior of linear VAEs with a fixed covariance matrix $\sigma^2 I_d$ and a varying $\beta_{\mathrm{VAE}}$, following the common practical approach in Higgins et al. (2016), to explore how the RD curve depends on the dataset size. As noted in (Rybkin et al., 2021), when $\sigma^2 = \beta_{\mathrm{VAE}}/2$, beta-VAE (Higgins et al., 2016) and $\sigma$-VAE are equivalent in optimization. Extending this analysis to cases where $\sigma$ is parametrized by learnable parameters, as in Rybkin et al. (2021), remains an important direction for future work. Note that, unlike the analysis of autoencoder (Nguyen, 2021), this study does not assume tied weights, i.e., $V^\top = W^\top$, which is a non-general constraint in VAEs.

**Training algorithm**    The VAE is trained by the following optimization problem:

$$(\hat{W}(\mathcal{D}), \hat{V}(\mathcal{D}), \hat{D}(\mathcal{D})) = \underset{W,V,D}{\text{argmin}}\, \mathcal{R}(W, V, D; \mathcal{D}, \beta_{\text{VAE}}, \lambda), \tag{6}$$

$$\mathcal{R}(W, V, D; \mathcal{D}, \beta_{\text{VAE}}, \lambda) \triangleq \sum_{\mu=1}^{n} \mathcal{L}(W, V, D; \boldsymbol{x}^{\mu}, \beta_{\text{VAE}}) + \lambda g(W, V), \tag{7}$$

where $\mathcal{L}(W, V, D; \boldsymbol{x}, \beta_{\text{VAE}})$ is defined by Eq. (1), and $g : \mathbb{R}^{d \times 2k} \to \mathbb{R}_+$ is an arbitrary convex regularizing function, corresponding to weight decay, which regulates the magnitudes of the parameters $W$ and $V$ with $\lambda \in \mathbb{R}_+$ being a regularization parameter. Many practitioners often include a weight decay term in VAE training (Kingma & Welling, 2013; Louizos et al., 2017). This study broadens the theoretical framework to cover these cases. Note that the following theoretical results are also applicable to scenarios without weight decay by setting $\lambda = 0$; see Appendix F.1.

**Evaluation metrics**    We use two evaluation metrics to investigate the behavior of linear VAEs. Following Lucas et al. (2019), we evaluate the rate to examine posterior collapse in the VAE:

$$R = \mathbb{E}_{p_{\mathcal{D}}} D_{\text{KL}}[q_{\hat{V},\hat{D}}(\boldsymbol{z}|\boldsymbol{x}) \| p(\boldsymbol{z})]. \tag{8}$$

We define posterior collapse as occurring when this rate equals zero, $R = 0$. This metric corresponds to the special case of the $(0, 0)$-collapsed condition discussed in Lucas et al. (2019). Further details on this correspondence are provided in Appendix C.

In addition, following the analysis of autoencoders (Refinetti & Goldt, 2022; Nguyen, 2021), we evaluate the signal recovery error to assess how well the decoder reconstructs the true distribution rather than focusing on the latent space. The signal recovery error is defined as

$$\varepsilon_g(W, W^*) = \frac{1}{d}\mathbb{E}_{p_{\mathcal{D}}}\mathbb{E}_{\boldsymbol{c}} \left\| \sqrt{\rho} \sum_{l^*=1}^{k^*} \boldsymbol{w}_{l^*} c_{l^*} - \sum_{l=1}^{k} \hat{\boldsymbol{w}}_l c_l \right\|^2. \tag{9}$$

where $\boldsymbol{w}_{l^*}$ and $\hat{\boldsymbol{w}}_l$ are column vectors of $W^*$ and $\hat{W}(\mathcal{D})$, respectively, and $\mathbb{E}_{\boldsymbol{c}}[\cdot]$ denotes the expectation over $p(\boldsymbol{c}) = \mathcal{N}\left(\boldsymbol{0}_{\max[k,k^*]}, I_{\max[k,k^*]}\right)$. The signal recovery error measures the extent of the signal recovery from the training data. Note that the distortion is defined as the squared error when data is encoded by the encoder $q_{V,D}$ and subsequently reconstructed by the decoder $p_W$, and is formally expressed as $\mathbb{E}_{p_{\mathcal{D}}}\mathbb{E}_{q_{V,D}}[-\log p_W(\boldsymbol{x}|\boldsymbol{z})]$. In contrast, the signal recovery error quantifies how closely the data generated by decoding latent variables sampled from a multivariate standard Gaussian distribution approximates the true distribution, rather than the compression performance.

**High-dimensional limit**    We analyze the optimization problem in Eq. (6) in the high-dimensional limit where the input dimension $d$ and number of training data $n$ simultaneously tend to infinity, while their ratio $\alpha = n/d = \Theta(d^0)$, referred to as the sample complexity. The hidden layer widths $k$ and $k^*$, the signal and noise level $\rho$ and $\eta$, are also assumed to remain $\Theta(d^0)$. This corresponds to a rich limit, where the number of VAE parameters is comparable to the number of samples, and the model cannot trivially fit or memorize the training dataset. Therefore, this limit allows us to study the effect of finite dataset-size dependence in the VAE.

## 5 ASYMPTOTIC FORMULAE

In this section, we show the main results of this study, namely the asymptotic formulae for linear VAEs trained with the objective function Eq. (7). These results are obtained by converting the optimization problem of Eq. (6) into an analysis of a corresponding Boltzmann measure, which is then analyzed using the replica method; For further details on the explanation and derivation, refer to Appendix D.

We discuss the main result in the high dimensional limit under the following assumption:

**Assumption 5.1** $g(W, V)$ *is $l_2$ regularizer, i.e.,* $g(W, V) = 1/2(\|W\|_F^2 + \|V\|_F^2)$.

Under this assumption, we present the main claim regarding the signal recovery error $\varepsilon_g$.

**Claim 5.2 (Asymptotics for VAE trained with Eq. (6))** *In the high-dimensional limit $d, n \to \infty$ with a fixed ratio $\alpha = n/d = \Theta(d^0)$, the signal recovery error $\varepsilon_g$ is given by*

$$\varepsilon_g = k^* \rho - 2 \sum_{l^*=1}^{k^*} \sum_{l=1}^{k} m_{ll^*} + \sum_{l=1}^{k} \sum_{s=1}^{k} q_{ls}, \tag{10}$$

*where we introduce the summary statistics:*

$$Q = (q_{ls}) = \lim_{d \to \infty} \mathbb{E}_{\mathcal{D}} \left[ \frac{1}{d} \hat{W}^\top \hat{W} \right], \quad m = (m_{ll^*}) = \lim_{d \to \infty} \mathbb{E}_{\mathcal{D}} \left[ \frac{1}{d} \hat{W}^\top W^* \right]. \tag{11}$$

*The summary statistics $Q$ and $m$ can be determined as solutions of the following extremum operation:*

$$f = \frac{1}{2} \underset{\substack{G, g, \psi \\ \hat{G}, \hat{g}, \hat{\psi}}}{\mathrm{extr}} \left\{ \mathrm{tr} \left[ g\hat{g} + 2\psi\hat{\psi} - G\hat{G} \right] - \mathrm{tr} \left[ (\hat{G} + \lambda)^{-1} \hat{g} \right] - \mathbf{1}_{k^*}^\top \hat{\psi}^\top (\hat{G} + \lambda)^{-1} \hat{\psi} \mathbf{1}_{k^*} \right.$$

$$\left. + \frac{\alpha}{\sigma^2} \left( \mathrm{tr} \left[ AG - \sqrt{\frac{\rho}{\eta}} \psi^\top B + (I_{2k} - Ag)^{-1} (AGA + BB^\top) g \right] + \sigma^2 \sum_{l=1}^{k} \log \frac{e(Q_{ll} + \beta_{\mathrm{VAE}})}{\beta_{\mathrm{VAE}}} \right) \right\}, \tag{12}$$

*where* extr *indicates taking the extremum with respect to $\Theta$ and*

$$G = \begin{pmatrix} Q & R \\ R & E \end{pmatrix}, \, g = \begin{pmatrix} \chi & \omega \\ \omega & \zeta \end{pmatrix}, \, \psi = \begin{pmatrix} m \\ b \end{pmatrix}, \, \hat{G} = \begin{pmatrix} \hat{Q} & \hat{R} \\ \hat{R} & \hat{E} \end{pmatrix}, \, \hat{g} = \begin{pmatrix} \hat{\chi} & \hat{\omega} \\ \hat{\omega} & \hat{\zeta} \end{pmatrix}, \, \hat{\psi} = \begin{pmatrix} \hat{m} \\ \hat{b} \end{pmatrix}$$

$$A = \eta \begin{pmatrix} \mathbf{0}_{k \times k} & I_k \\ I_k & -(Q + \sigma^2 \beta_{\mathrm{VAE}} I_k) \end{pmatrix}, \, B = \sqrt{\rho\eta} \begin{pmatrix} -b \\ -m + (Q + \sigma^2 \beta_{\mathrm{VAE}} I_k)b \end{pmatrix}.$$

The summary statistics $m$ corresponds to the overlap of the signal $W^*$ and decoder parameter $W$; while $m_{ll^*} \neq 0$ indicates that the VAE recovers the signal $\boldsymbol{w}_{l^*}$, when $m_{ll^*} = 0$, the VAE does not learn the signal $\boldsymbol{w}_{l^*}$. The summary statistics $Q$ represents the norm of the decoder weights $W$, which measures the freedom of the parameter; a smaller $Q$ indicates a stronger regularization, yielding a smaller effective feasible region of the parameter (and vice versa). Additionally, the rate $R$ and distortion $D$ can be evaluated through these summary statistics.

**Claim 5.3** *In the high-dimensional limit $d, n \to \infty$, the rate $R(\hat{V}, \hat{D})$ and distortion $D(\hat{W}, \hat{V}, \hat{D})$ are also expressed as functions of $G$, $g$, and $\psi$, determined by the extremum problem Eq. (12).*

The details are in Appendix D. Claim 5.2 provides the asymptotic properties of the model at the global optimum of the objective function in Eq. (6). Eq. (12) provides the summary statistics Eq. (11), derived from the solutions of the low-dimensional optimization problem in Eq. (12). The high-dimensional optimization problem Eq. (6) and the high-dimensional average over the training dataset $\mathcal{D}$ are reduced to a simpler tractable system of optimization problem over $2k(8k + 2k^*)$ variables Eq. (15) in Appendix D, which can be easily solved numerically. It is important to note that all the summary statistics involved in Eq. (12) are finite-dimensional as $d, n \to +\infty$, meaning that Claim 5.2 provides a fully asymptotic characterization, as it does not involve any high-dimensional variables. Finally, let us stress once more that the replica method employed in the derivation of these results should be viewed as a strong heuristic but does not constitute rigorous proof; thus, the results are presented here as a claim. Furthermore, Assumption 5.1 can be relaxed to address arbitrary convex regularizer $g(\cdot, \cdot)$, but the free energy becomes more intricate formulae. For this reason, $l_2$ regularizer is chosen.

# 6 RESULTS

We now analyze how the signal recovery error $\varepsilon_g$ and RD curve are influenced by $\alpha$ and $\beta_{\mathrm{VAE}}$ using Claim 5.2. While Claim 5.2 is stated in full generality, for definiteness in the rest of the manuscript, we focus on a minimal setting $k = 1$ and $k^* = 1$ to comprehend posterior collapse. This minimal setting is found to already display meaningful results even for more realistic datasets and complicated

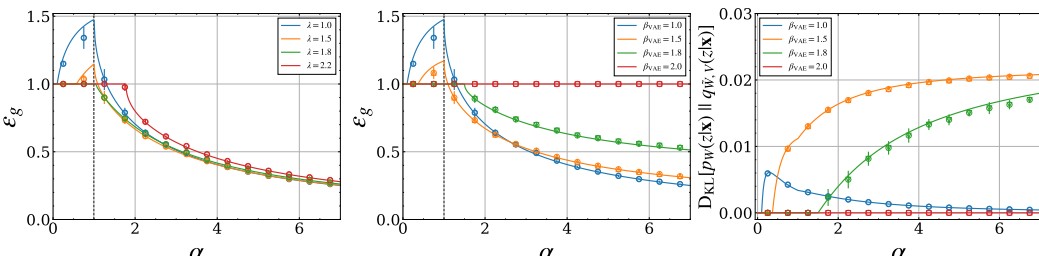

Figure 2: (Left) signal recovery error as a function of sample complexity $\alpha$ for fixed parameters $\beta_{\text{VAE}} = 1$ and varying $\lambda$. (Middle) signal recovery error for different $\beta_{\text{VAE}}$ with fixed parameter $\lambda = 1$. (Right) KL divergence between the true and variational posterior with fixed parameter $\lambda = 1$ for different $\beta_{\text{VAE}}$. Each data point in all the plots represents the average result of five different numerical simulations at $d = 5,000$ using gradient descent; the error bars represent the standard deviations of the results.

non-linear VAE, as discussed in Section 6.5. We leave the thorough exploration of settings with $k > 1$ and $k^* > 1$ for future work. When $\sigma^2$ is not a learnable parameter, adjusting $\beta_{\text{VAE}}$ while keeping $\sigma^2$ fixed is equivalent to adjusting $\sigma^2$ while keeping $\beta_{\text{VAE}}$ fixed are equivalent in optimization. Thus, we fix $\sigma^2 = 1$ and focus on investigating the dependence on $\beta_{\text{VAE}}$. In addition, numerical experiments are conducted to verify the consistency of our theory, which are implemented with `Pytorch` of Adam optimizer (Kingma & Ba, 2014).

### 6.1 LEARNING CURVE OF SIGNAL RECOVERY ERROR

First, we clarify the relationship between the signal recovery error and $\beta_{\text{VAE}}$. The signal recovery error and KL divergence $D_{\text{KL}}[p_{\hat{W}}(z|\boldsymbol{x}) \| q_{\hat{V}, \hat{D}}(z|\boldsymbol{x})]$ evaluated from the solutions of the optimization problem of Eq. (6) are plotted as the solid lines in Fig. 2 and compared with the numerical simulations for $l_2$ regularization weight $\lambda = 1.0$. The agreement between the theory and simulations is compelling. Our results can be summarized in three points as follows. In addition, the dependence of signal recovery error on $\beta_{\text{VAE}}$ and $\alpha$ without weight decay, i.e., $\lambda = 0$, is shown in the Appendix F.1. In this case, the results are qualitatively similar to those described below.

**Interpolation peak as in supervised learning**  We demonstrate that the well-known interpolation peak in supervised regression (Mignacco et al., 2020; Hastie et al., 2022; Opper & Kinzel, 1996) also occurs in VAEs in unsupervised scenarios. The interpolation peak in the supervised regression had a characteristic peak in the signal recovery error at $\alpha = 1$ with a small ridge regularization parameter, and the peak gradually decreased as the regularization parameter increased. Fig. 2 demonstrates the dependence of the signal recovery error $\varepsilon_g$ obtained by the replica method on $\beta_{\text{VAE}}$ and $\lambda$, together with the numerical experimental results with a finite dataset size. The curves for small $\beta_{\text{VAE}}$ and $\lambda$ values show a peak at $\alpha = 1$. This peak tends to disappear smoothly as the increasing $\beta_{\text{VAE}}$ and $\lambda$. This implies that the peak is a universal phenomenon observed in both supervised and unsupervised settings.

**Long plateau in $\varepsilon_g$ with a large beta**  The middle panel of Fig. 2 demonstrates the $\alpha$-dependence of the signal recovery error $\varepsilon_g$ for various $\beta_{\text{VAE}}$. For a smaller $\beta_{\text{VAE}}$, the signal recovery error $\varepsilon_g$ begins decreasing from $\alpha = 1$. Meanwhile, as $\beta_{\text{VAE}}$ increased, a long plateau appears in the range of $\alpha$ before the curve begins to decrease. Notably, the length of this plateau increases with increasing $\beta_{\text{VAE}}$. Moreover, when the value of $\beta_{\text{VAE}}$ exceeds 2, the decrease in the signal recovery error $\varepsilon_g$ disappears completely. The exact points at which the signal recovery error begins to decrease and remains 1 are explained in the following section, with a corresponding description of the phase diagram.

**Optimal beta depends on sample complexity**  We clarify that the optimal value of $\beta_{\text{VAE}}$ that minimizes the signal recovery error $\varepsilon_g$ depends on $\alpha$. Specifically, in the smaller $\alpha$ regime ranging from approximately $\alpha = 1$ to $\alpha \approx 2.6$, the signal recovery error $\varepsilon_g$ is minimized by $\beta_{\text{VAE}} =$

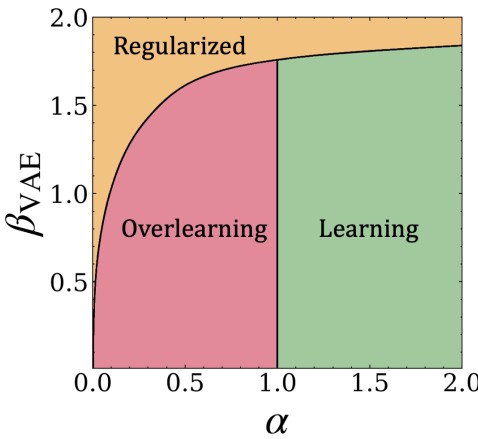
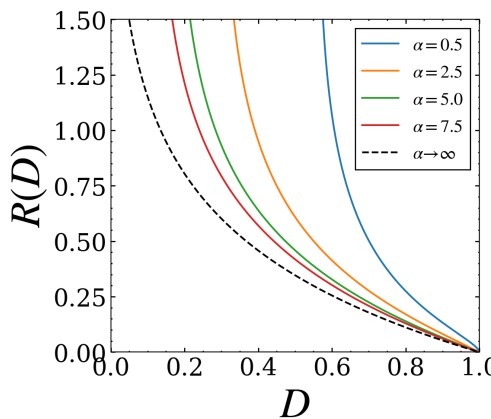

Figure 3: Phase diagram for $\lambda = 1$: Learning phase, overfitting phase, and regularized phase.

Figure 4: RD curve for $\lambda = 1$ with various values of $\alpha$.

1.5. However, in the larger $\alpha$ regime, the optimal value is $\beta_{\mathrm{VAE}} = 1$. In addition, the right panel of Fig. 2 presents the KL divergence between the true posterior and the variational posterior, $D_{\mathrm{KL}}[p_W(z|\boldsymbol{x}) \| q_{V,D}(z|\boldsymbol{x})]$, as a function of $\alpha$ for different values of $\beta_{\mathrm{VAE}}$. The figure demonstrates that minimizing the signal recovery error $\varepsilon_g$ does not necessarily bring the true posterior $p_W(z|\boldsymbol{x})$ closer to the variational posterior $q_{V,D}(z|\boldsymbol{x})$. In fact, despite the signal recovery error $\varepsilon_g$ being minimized at $\beta_{\mathrm{VAE}} = 1.5$, the KL divergence for the $\beta_{\mathrm{VAE}}$ is not minimal in the range between $\alpha = 1$ and $\alpha \approx 2.6$. Furthermore, unlike the strength of ridge regularization, $\lambda$, an improperly chosen $\beta_{\mathrm{VAE}}$ for a given $\alpha$ can result in significant performance variations. Therefore, $\beta_{\mathrm{VAE}}$ must be carefully optimized for each specific value of $\alpha$. This observation offers a crucial insight for practitioners of VAEs in engineering applications.

## 6.2 Phase diagram

Based on the extreme values of summary statistics $m$ and $Q$ in Eq. (12), we next discuss the phase diagram in terms of $\beta_{\mathrm{VAE}}$. The following three distinct phases are identified, as shown in Fig. 3:

- Learning phase (green region, $m \neq 0, Q \neq 0, R \neq 0$): The VAE recovers the signal and avoids posterior collapse.

- overfitting phase (red region, $m = 0, Q \neq 0, R = 0$): The effects of the rate and ridge regularizations are small, causing overfitting of the background noise in the data.

- Regularized phase (orange region, $m = 0, Q = 0, R = 0$): The rate and ridge regularizations restrict the degrees of freedom of the learnable parameters, leading to posterior collapse.

As noted in the previous section, the boundaries between the overfitting and learning phases, as well as those between the regularized and learning phases in the phase diagram, precisely correspond to the point where the signal recovery error begins to decrease. The phase diagram shows that as $\beta_{\mathrm{VAE}}$ increases, the transition to the learning phase becomes more challenging, even with a sufficiently large $\alpha$, indicating the *long plateau* described above.

## 6.3 Large dataset limit

The phase diagram presented in the previous section does not provide information on whether it is possible to reach the learning phase by increasing $\alpha$ for any $\beta_{\mathrm{VAE}}$. This feasibility is demonstrated by an analysis in the large $\alpha$ limit. Furthermore, the optimal value of $\beta_{\mathrm{VAE}}$ that minimizes the signal recovery error in the large $\alpha$ limit is derived. First, we present the following claim:

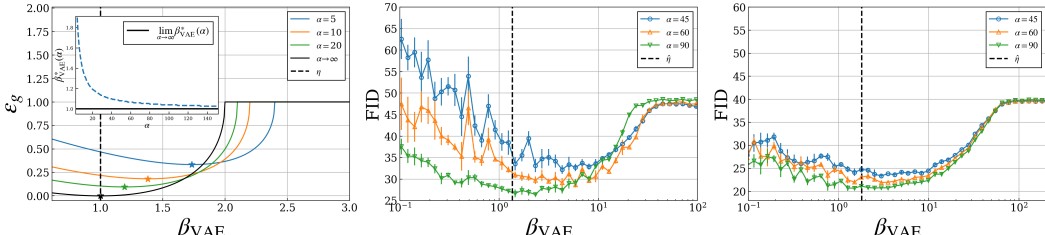

Figure 5: (Left) $\beta_{\text{VAE}}$ dependence of the signal recovery error $\varepsilon_g$ predicted by Claim 5.2 in linear VAE. The inset shows the $\alpha$-dependence of the optimal $\beta_{\text{VAE}}^*$. FIDs as a function of $\beta_{\text{VAE}}$ for the MNIST dataset (Middle) and FashionMNIST (Right) with a nonlinear VAE. Dashed vertical lines indicate the estimated noise strength $\hat{\eta}$. The error bars represent the standard deviations of the results.

**Claim 6.1** *In a large $\alpha$ limit and for any $\lambda$, when $\beta_{\text{VAE}} < \rho + \eta$, the summary statistics $m$ and the signal recovery error $\varepsilon_g$ are expressed as follows:*

$$R = \frac{1}{2} \log\left(\frac{\eta + \rho}{\beta_{\text{VAE}}}\right), \quad \varepsilon_g = \rho - \sqrt{\eta + \rho - \beta_{\text{VAE}}}(2\sqrt{\rho} - \sqrt{\eta + \rho - \beta_{\text{VAE}}}),$$

*respectively, and when $\beta_{\text{VAE}} \geq \rho + \eta$, $R = 0$ and $\varepsilon_g = \rho$, indicating that posterior collapse occurs.*

Based on Claim 6.1, once $\beta_{\text{VAE}}$ exceeds the threshold $\hat{\beta}_{\text{VAE}} = \rho + \eta$, the learning phase cannot be reached despite increasing $\alpha$, which indicates that the posterior collapse is inevitable. This result suggests that $\beta_{\text{VAE}}$ can be a risky parameter and that learning can fail regardless of the dataset size. Furthermore, the extremum calculations of the signal recovery error in Claim 6.1 demonstrate that the signal recovery error reaches a minimum value at $\beta_{\text{VAE}} = \eta$, which implies that the optimal result is achieved when $\beta_{\text{VAE}}$ equals the background noise strength $\eta$. Additionally, Claim 6.1 can be extended to any $k = k^* = \mathcal{O}(d^0)$ under certain assumptions, showing that posterior collapse consistently occurs at the threshold $\beta_{\text{VAE}} = \rho + \eta$, regardless of the size of the dataset. Therefore, this result remains robust even when some latent variables exist. The detailed proof can be found in Appendix D.4.

### 6.4 RD CURVE

We demonstrate that the RD curve in the large $\alpha$ limit is as follows.

**Claim 6.2** *In a large $\alpha$ limit, the RD curve $R$ of the linear VAE equals that of a Gaussian source (Cover, 1999) for any $\lambda \in \mathbb{R}_+$:*

$$R \triangleq \mathbb{E}_{p_{\mathcal{D}}} R(\hat{V}, \hat{D}) = \begin{cases} \frac{1}{2} \log \frac{\eta + \rho}{2D} & 0 \leq D < \frac{\rho + \eta}{2}, \\ 0 & D \geq \frac{\rho + \eta}{2}, \end{cases}$$

$$D \triangleq \mathbb{E}_{p_{\mathcal{D}}} D(\hat{W}, \hat{V}, \hat{D}) = \begin{cases} \frac{\beta_{\text{VAE}}}{2} & 0 \leq \beta_{\text{VAE}} < \rho + \eta, \\ \frac{\rho + \eta}{2} & \beta_{\text{VAE}} \geq \rho + \eta. \end{cases}$$

The detailed derivation can be found in Appendix D.5, and a brief explanation of the RD function for the Gaussian source is provided in Appendix B. Claim 6.2 suggests that the VAE achieves an optimal compression rate in a large $\alpha$ limit. Furthermore, the rate introduced by Alemi et al. (2018) is found to coincide with the rate of discrete quantization of the RD theorem (Cover, 1999) in the large $\alpha$ limit, indicating that the rate is a truly generalized form of the rate of the discrete quantization in the RD theory. Fig. 4 shows the RD curve for both the large $\alpha$ limit and finite $\alpha$, demonstrating that a relatively large dataset is required to achieve the optimal RD curve in the high-rate and low-distortion regions. Moreover, when $dR(D)/dD = -1$, the VAE achieves an optimal signal recovery error with $\beta_{\text{VAE}} = \eta$. In Appendix F.1, we also show that this property of the RD curve is consistent for VAEs without weight decay, i.e., $\lambda = 0$.

### 6.5 ROBUSTNESS OF REPLICA PREDICTION AGAINST REAL DATA

It is reasonable to question whether the theoretical analysis can explain the phenomena observed in more complex real-world datasets with nonlinear VAEs. The answer is empirically positive, as

described below. Specifically, we investigate whether the existence of the posterior collapse threshold and the dependency of generalization performance on $\beta_{\text{VAE}}$ and $\alpha$ predicted by Claim 5.2 in the Linear VAE, remain consistent when applied to real-world datasets with nonlinear VAEs. We compare the generalization properties predicted by the theoretical analysis with those observed in Fashion MNIST (Deng, 2012) and MNIST (Deng, 2012) using a 3-layer MLP for the encoder and decoder. For these datasets, we calculated $\beta_{\text{VAE}}$ dependence of Fréchet Inception Distance (FID) (Heusel et al., 2017), one of the most widely used generalization metrics for generated images, instead of the signal recovery error in Eq. (9). Here, $\hat{\eta}$ represents a noise strength in Eq. (4), estimated by the empirical standard deviation of the bulk, consisting of the eigenmodes of the empirical covariance matrix, under an $80\%$ cumulative contribution rate. The result remains consistent even when the rate is set to $90\%$ or $70\%$. Details of the experimental settings can be found in Appendix E.1.

Fig. 5 shows that the FID values for both Fashion MNIST and MNIST qualitatively match the theoretical predictions. Inevitable posterior collapse occurs as $\beta_{\text{VAE}}$ increases, and the threshold shifts towards higher $\beta_{\text{VAE}}$ as the sample complexity $\alpha$ increases, which is consistent with the theoretical results. Additionally, the optimal $\beta_{\text{VAE}}^*$ approaches the estimated value $\hat{\eta}$ as $\alpha$ increases. The correction from the optimal $\lim_{\alpha \to \infty} \beta_{\text{VAE}}^*(\alpha)$ is positive in the direction of $\beta_{\text{VAE}}$, which is also consistent with theoretical results. These observations suggest that the generalization behavior of real datasets is well captured by the SCM model, indicating the presence of Gaussian universality (Hu & Lu, 2022; Montanari & Saeed, 2022; Loureiro et al., 2021). This opens new avenues for future research, as Gaussian universality has been explored in classification and regression. The qualitative behavior remains consistent when applied to the CIFAR10 dataset (Krizhevsky, 2009), which consists of color images and a convolutional neural network (CNN). The experimental results are provided in Appendix F.2.

# 7 CONCLUSION

We provide a high-dimensional asymptotic characterization of trained linear VAEs, clarifying the relationship between dataset size, $\beta_{\text{VAE}}$, posterior collapse, and RD curve. Specifically, these results show an "inevitable posterior collapse" regardless of the dataset size beyond a certain beta threshold and the dataset-size dependence of the RD curve, indicating that relatively large datasets are required in high-rate regions. These findings also explain the qualitative behavior for realistic datasets and nonlinear VAEs, providing theoretical insights that support longstanding practical intuitions about VAEs.

Finally, building on our analysis, we present insights for the engineering applications of VAEs. This study reveals that the parameter $\beta_{\text{VAE}}$, unlike the conventional ridge regularization coefficient $\lambda$, requires careful tuning based on dataset size. Inappropriate tuning leads to significant degradation in generalization performance. In particular, an excessively large $\beta_{\text{VAE}}$ induces a "plateau phenomenon" that persists despite increases in dataset size, hindering further performance improvements and eventually causing inevitable posterior collapse. These findings underscore that $\beta_{\text{VAE}}$ is a highly sensitive and potentially risky parameter requiring meticulous adjustment. This study also reveals that in the limit of large dataset sizes, the optimal value of $\beta_{\text{VAE}}$ corresponds to the strength of background noise in the data. In contrast, for finite datasets, the optimal value of $\beta_{\text{VAE}}$ tends to shift to higher values. This tendency consistently holds in our numerical experiments across real-world datasets and VAEs with nonlinear structures, demonstrating its robustness. This directional adjustment offers a critical guideline for effectively tuning $\beta_{\text{VAE}}$. By quantitatively examining the conventional claim that "a large $\beta_{\text{VAE}}$ induces posterior collapse" through a minimal model based on a linear VAE, we not only clarified the underlying mechanism but also provided practical guidelines for parameter tuning.

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

## A  OVERVIEW

This supplementary material provides extended explanations, implementation details, and additional results for the paper *High-dimensional Asymptotics of VAEs: Threshold of Posterior Collapse and Dataset-Size Dependence of Rate-Distortion Curve*.

## B   REVIEW OF RATE DISTORTION THEORY

The rate-distortion theory was introduced by Shannon et al. (1959) and then further developed by Berger et al. (1975); Berger & Gibson (1998). This theoretical framework describes the minimum bit rate (rate) required for encoding a source, subject to a given distortion measure. In recent years, it has been used to understand machine learning (Gao et al., 2019; Alemi et al., 2018; Theis et al., 2022; Brekelmans et al., 2019; Isik et al., 2022).

Let $X^P = \{X_1, \ldots, X_P\} \in \mathcal{X}^P$ be i.i.d random variables from the distribution $P(x)$. An encoder $f_P : \mathcal{X}^P \to \{1, 2, \ldots, 2^{P \times R}\}$ maps the input $X^P$ into a quantized vector, and a decoder $g_P : \{1, 2, \ldots, 2^{P \times R}\} \to \mathcal{X}^P$ reconstructs the input by a decoded input $\hat{X}^P$ from the quantized vector. To measure the discrepancy between the original and decoded inputs, a distortion function $d : \mathcal{X} \times \mathcal{X} \to \mathbb{R}_+$ is introduced. The distortion for the input $X^P$ and decoded input $\hat{X}^P$ is defined as the average distortion between each pair $X_i$ and $\hat{X}_i$. Commonly used distortion functions are the Hamming distortion function defined as $d(x, \hat{x}) = \mathbb{I}[x \neq \hat{x}]$ for $X = \{0, 1\}$ where $\mathbb{I}$ is the indicator function, and the squared error distortion function defined as $d(x, \hat{x}) = (x - \hat{x})^2$ for $X = \mathbb{R}$. We are ready to define the RD function.

**Definition B.1** *A rate-distortion pair $(R, D)$ is achievable if there exists a (probabilistic) encoder-decoder $(f_P, g_P)$ such that the quantized vector has size $2^{P \times R}$ and the expected distortion $\lim_{P \to \infty} [d(X^P, g_P(f_P(X^P)))] \leq D$.*

**Definition B.2** *The RD function $R(D)$ is the infimum of rates $R$ such that the RD pair $(R, D)$ is achievable.*

The main theorem of the RD theory (Cover, 1999) states as follows,

**Theorem B.3** *Given an upper bound of distortion $D$, the following equation holds:*

$$R(D) = \min_{P(\hat{X}|X):\mathbb{E}[d(X,\hat{X})] \leq D} I(X; \hat{X}) \tag{13}$$

The RD theorem provides the fundamental limit of data compression, i.e., how many minimum bits are needed to compress the input, given the quality of the reconstructed input.

### B.1   RD OF GAUSSIAN SOURCE.

We give an example of the RD function for Gaussian input.

**Proposition B.4** *If $X \sim \mathcal{N}(0, \sigma^2)$, the RD function is given by*

$$R(D) = \begin{cases} \frac{1}{2} \log_2 \frac{\sigma^2}{D} & D \leq \sigma^2 \\ 0 & D > \sigma^2 \end{cases}.$$

If the required distortion is larger than the variance of the Gaussian variable $\sigma^2$, we simply transmit $\hat{X} = 0$; otherwise, we transmit $\hat{X}$ such that $\hat{X} \sim \mathcal{N}(0, \sigma^2 - D)$, $X - \hat{X} \sim \mathcal{N}(0, D)$ where $\hat{X}$ and $X - \hat{X}$ are independent.

## C   EVALUATION METRIC OF POSTERIOR COLLAPSE

To evaluate the degree of posterior collapse, Lucas et al. (2019) defined a latent variable dimension $z_i$ as being $(\epsilon, \delta)$-collapsed if it satisfies $\mathbb{P}_{\mathcal{D}}[D_{\mathrm{KL}}(q_{\hat{V}, \hat{D}}(z_i|\mathbf{x})\|p(z_i)) < \varepsilon] \geq 1 - \delta$. While this can also be evaluated using the summary statistic in Claim 5.2, for simplicity, we consider posterior collapse to occur when $R = \sum_i \mathbb{E}_{\mathcal{D}}[D_{\mathrm{KL}}(q_{\hat{V}, \hat{D}}(z_i|\mathbf{x})\|p(z_i))] = 0$. As $\delta \to 0$ and $\varepsilon \to 0$, this implies that almost surely under $p_{\mathcal{D}}$, $D_{\mathrm{KL}}(q_{\hat{V}, \hat{D}}(z_i|\mathbf{x})\|p(z_i)) = 0$, leading to $\mathbb{E}_{p_{\mathcal{D}}}[D_{\mathrm{KL}}(q_{\hat{V}, \hat{D}}(z_i|\mathbf{x})\|p(z_i))] = 0$. Therefore, our definition of $R = \sum_i \mathbb{E}_{\mathcal{D}}[D_{\mathrm{KL}}(q_{\hat{V}, \hat{D}}(z_i|\mathbf{x})\|p(z_i))]$ is consistent with all latent variables $\mathbf{z}$ being $(0, 0)$-collapsed.

# D DERIVATION OF CLAIMS

Here, we present the detailed derivation of Claims 5.2, 5.3, 6.1, 6.2.

## D.1 REPLICA FORMULATION

The Boltzmann distribution is defined as follows:

$$p(W, V, D; \mathcal{D}, \gamma) \triangleq \frac{1}{Z(\mathcal{D}, \gamma)} e^{-\gamma \mathcal{R}(W, V, D; \mathcal{D}, \beta_{\mathrm{VAE}}, \lambda)} \tag{14}$$

where $Z(\mathcal{D}, \gamma)$ is the normalization constant known as the partition function in statistical mechanics. Note that in the limit $\gamma \to \infty$, Eq. (14) converges to a distribution concentrated on the $(\hat{W}(\mathcal{D}), \hat{V}(\mathcal{D}), \hat{D}(\mathcal{D}))$. Thus, the expectation of any function $\psi(\hat{W}(\mathcal{D}), \hat{V}(\mathcal{D}), \hat{D}(\mathcal{D}))$, which includes signal recovery error $\varepsilon_g$, rate and distortion, over the dataset can be expressed as an average over a limiting distribution as follows:

$$\mathbb{E}_{\mathcal{D}} \psi(\hat{W}(\mathcal{D}), \hat{V}(\mathcal{D}), \hat{D}(\mathcal{D})) = \lim_{\gamma \to \infty} \mathbb{E}_{\mathcal{D}} \int dW \, dV \, dD \, \psi(W, V, D) p(W, V, D; \mathcal{D}, \gamma).$$

The idea of the replica method (Mézard et al., 1987; Mezard & Montanari, 2009; Edwards & Anderson, 1975) is to compute the moment generating function (also known as the free-energy density) as follows:

$$f = -\lim_{\gamma \to \infty} \frac{1}{\gamma d} \mathbb{E}_{\mathcal{D}} \log Z(\mathcal{D}, \gamma). \tag{15}$$

Although Eq. (15) is difficult to calculate in a straightforward manner, this can be resolved by using the replica method (Mézard et al., 1987; Mezard & Montanari, 2009; Edwards & Anderson, 1975), which is based on the following equality:

$$\mathbb{E}_{\mathcal{D}} \log Z(\mathcal{D}, \gamma) = \lim_{p \to +0} \frac{\log \mathbb{E}_{\mathcal{D}} Z^p(\mathcal{D}, \gamma)}{p}. \tag{16}$$

Instead of directly handling the cumbersome log expression in Eq. (15), we can calculate the average of the $n$-th power of $Z(\mathcal{D}, \gamma)$ for $p \in \mathbb{N}$, analytically continue this expression to $p \in \mathbb{R}$, and finally takes the limit $p \to +0$. Based on this replica trick, it is sufficient to calculate the following:

$$\mathbb{E}_{\mathcal{D}} Z^p(\mathcal{D}, \gamma) = \mathbb{E}_{\mathcal{D}} \int \prod_{\nu=1}^{p} dW^\nu dV^\nu dD^\nu \prod_{\nu=1}^{p} e^{-\gamma \mathcal{R}(W^\nu, V^\nu, D^\nu; \mathcal{D}, \beta_{\mathrm{VAE}}, \lambda)} \tag{17}$$

up to the first order of $p$ to take the $p \to +0$ limit on the right-hand side of Eq. (16).

## D.2 REPLICATED PARTITION FUNCTION

To calculate free-energy density, it is sufficient to calculate the replicated partition function, as mentioned in Section 4.1. The replicated partition function is expressed as

$$\mathbb{E}_{\mathcal{D}} Z^p(\mathcal{D}, \gamma)$$

$$= \mathbb{E}_{\mathcal{D}} \int \prod_{\nu=1}^{p} dW^\nu dV^\nu dD^\nu \prod_{\nu=1}^{p} e^{-\gamma \mathcal{R}(W^\nu, V^\nu, D^\nu; \mathcal{D}, \beta_{\mathrm{VAE}}, \lambda)}$$

$$= \mathbb{E}_{\mathcal{D}} \int \prod_{\nu=1}^{p} dW^\nu dV^\nu dD^\nu e^{-\frac{\gamma\lambda}{2} \sum_{\nu=1}^{p} \left( \|W^\nu\|_F^2 + \|V^\nu\|_F^2 \right)} \prod_{\nu=1}^{p} \left( e^{-\gamma \sum_{\mu=1}^{n} \mathcal{L}(W^\nu, V^\nu, D^\nu; \boldsymbol{x}^\mu, \beta_{\mathrm{VAE}})} \right)$$

$$= \int \prod_{\nu=1}^{p} dW^\nu dV^\nu dD^\nu e^{-\frac{\gamma\lambda}{2} \sum_{\nu=1}^{p} \left( \|W^\nu\|_F^2 + \|V^\nu\|_F^2 \right)} \prod_{\mu=1}^{n} \mathbb{E}_{\boldsymbol{c}^\mu, \boldsymbol{n}^\mu} \prod_{\nu=1}^{p} e^{-\gamma \sum_{\mu=1}^{n} \mathcal{L}(W^\nu, V^\nu, D^\nu; \boldsymbol{c}^\mu, \boldsymbol{n}^\mu \beta_{\mathrm{VAE}})}$$

$$= \int \prod_{\nu=1}^{p} dW^\nu dV^\nu dD^\nu e^{-\frac{\gamma\lambda}{2} \sum_{\nu=1}^{p} \left( \|W^\nu\|_F^2 + \|V^\nu\|_F^2 \right)} \left( \mathbb{E}_{\boldsymbol{c}, \boldsymbol{n}} \left[ e^{-\gamma \sum_{\nu=1}^{p} \mathcal{L}(W^\nu, V^\nu, D^\nu; \boldsymbol{c}, \boldsymbol{n}, \beta_{\mathrm{VAE}})} \right] \right)^n,$$

where $\mathcal{L}(W^\nu, V^\nu, D^\nu; \boldsymbol{c}, \boldsymbol{n}, \beta_{\text{VAE}})$ is given by

$$
\mathcal{L}(W^\nu, V^\nu, D^\nu; \boldsymbol{c}, \boldsymbol{n}, \beta_{\text{VAE}})
$$

$$
= \frac{1}{2\sigma^2}\left( \left\| \sqrt{\frac{\rho}{d}}W^*\boldsymbol{c} + \sqrt{\eta}\boldsymbol{n} \right\|^2 - 2\left( \frac{\sqrt{\rho}}{d}(W^\nu)^\top W^*\boldsymbol{c} + \sqrt{\frac{\eta}{d}}(W^\nu)^\top \boldsymbol{n} \right)^\top \left( \frac{\sqrt{\rho}}{d}(V^\nu)^\top W^*\boldsymbol{c} + \sqrt{\frac{\eta}{d}}(V^\nu)^\top \boldsymbol{n} \right) \right.
$$

$$
+ \left( \frac{\sqrt{\rho}}{d}(V^\nu)^\top W^*\boldsymbol{c} + \sqrt{\frac{\eta}{d}}(V^\nu)^\top \boldsymbol{n} \right)^\top \frac{(W^\nu)^\top W^\nu}{d} \left( \frac{\sqrt{\rho}}{d}(V^\nu)^\top W^*\boldsymbol{c} + \sqrt{\frac{\eta}{d}}(V^\nu)^\top \boldsymbol{n} \right) + \frac{1}{d}(W^\nu)^\top W^\nu D^\nu
$$

$$
\left. + \beta_{\text{VAE}}\left( \left\| \frac{\sqrt{\rho}}{d}(V^\nu)^\top W^*\boldsymbol{c} + \sqrt{\frac{\eta}{d}}(V^\nu)^\top \boldsymbol{n} \right\|^2 + \text{tr}(D^\nu) - \text{tr}(\log D^\nu) \right) \right).
$$

To perform the average over $\boldsymbol{n}$, we notice that, since $\boldsymbol{n}$ follows a multivariate normal distribution $\mathcal{N}(\boldsymbol{0}_d, I_d)$, $\boldsymbol{h} \triangleq \oplus_{\nu=1}^p (\boldsymbol{u}^\nu \oplus \tilde{\boldsymbol{u}}^\nu) \in \mathbb{R}^{2kd}$ with

$$
\boldsymbol{u}^\nu \triangleq \frac{1}{\sqrt{d}}(W^\nu)^\top \boldsymbol{n}^\mu \in \mathbb{R}^k, \ \tilde{\boldsymbol{u}}^\nu \triangleq \frac{1}{\sqrt{d}}(V^\nu)^\top \boldsymbol{n}^\mu \in \mathbb{R}^k
$$

follows a Gaussian multivariate distribution, $p(\boldsymbol{h}) = \mathcal{N}(\boldsymbol{h}; \boldsymbol{0}_{2kp}, \Sigma)$, where

$$
\mathbb{E}_{\boldsymbol{n}}\boldsymbol{h}(\boldsymbol{h})^\top = \Sigma, \ \Sigma^{\nu\kappa} = \begin{pmatrix} Q^{\nu\kappa} & R^{\nu\kappa} \\ R^{\nu\kappa} & E^{\nu\kappa} \end{pmatrix}, \ Q^{\nu\kappa} = \frac{1}{d}(W^\nu)^\top W^\kappa, \ E^{\nu\kappa} = \frac{1}{d}(V^\nu)^\top V^\kappa, \ R^{\nu\kappa} = \frac{1}{d}(W^\nu)^\top V^\kappa.
$$

By introducing the auxiliary variables through the trivial identities as follows:

$$
1 = \prod_{(\nu,l);(\kappa,s)} d \int \delta\left( Q_{ls}^{\nu\kappa}d - (\boldsymbol{w}_l^\nu)^\top \boldsymbol{w}_s^\kappa \right) dQ,
$$

$$
1 = \prod_{(\nu,l);(\kappa,s)} d \int \delta\left( E_{ls}^{\nu\kappa}d - (\boldsymbol{v}_l^\nu)^\top \boldsymbol{v}_s^\kappa \right) dE,
$$

$$
1 = \prod_{(\nu,l);(\kappa,s)} d \int \delta\left( R_{ls}^{\nu\kappa}d - (\boldsymbol{w}_l^\nu)^\top \boldsymbol{v}_s^\kappa \right) dR,
$$

$$
1 = \prod_{(\nu,s);(\nu,l^*)} d \int \delta\left( m_{sl^*}^\nu d - (\boldsymbol{w}_s^\nu)^\top \boldsymbol{w}_{l^*}^* \right) dm,
$$

$$
1 = \prod_{(\nu,s);(\nu,l^*)} d \int \delta\left( b_{sl^*}^\nu d - (\boldsymbol{v}_s^\nu)^\top \boldsymbol{w}_{l^*}^* \right) db,
$$

the replicated partition function is further expressed as

$$
\mathbb{E}_{\mathcal{D}}Z^p(\mathcal{D}, \gamma) = \int dQ dE dR dm db \left( \mathcal{S} \times \mathcal{E} \right),
$$

$$
\mathcal{S} \triangleq \int \prod_{\nu=1}^p dW^\nu dV^\nu \prod_{\nu,\kappa} \prod_{s,l} d\delta\left( Q_{sl}^{\nu\kappa}d - (\boldsymbol{w}_s^\nu)^\top \boldsymbol{w}_l^\kappa \right) d\delta\left( E_{sl}^{\nu\kappa}d - (\boldsymbol{v}_s^\nu)^\top \boldsymbol{v}_l^\kappa \right) d\delta\left( R_{sl}^{\nu\kappa}d - (\boldsymbol{w}_s^\nu)^\top \boldsymbol{v}_l^\kappa \right)
$$

$$
\prod_\nu \prod_{s,l} d\delta\left( m_{sl^*}^\nu d - (\boldsymbol{w}_s^\nu)^\top \boldsymbol{w}_{l^*}^* \right) d\delta\left( b_{sl^*}^\nu - (\boldsymbol{v}_s^\nu)^\top \boldsymbol{w}_l^* \right) \times e^{-\frac{\gamma\lambda}{2}\sum_\nu (\|W^\nu\|_F^2 + \|V^\nu\|_F^2)},
$$

$$
\mathcal{E} \triangleq \int \prod_\nu dD^\nu \left( \int D\boldsymbol{c} \int d\boldsymbol{h} \mathcal{N}(\boldsymbol{h}, \boldsymbol{0}_{2kp}, \Sigma) \times e^{-\gamma \sum_\nu \mathcal{L}(Q,E,R,m,d;\boldsymbol{h},\boldsymbol{c},\beta_{\text{VAE}},\lambda)} \right)^n,
$$

where $\boldsymbol{w}_{l^*}^*$, $\boldsymbol{w}_l^\nu$ and $\boldsymbol{v}_l^\nu$ are column vectors of $W^*$, $W^\nu$, and $V^\nu$, respectively. Assuming the replica symmetric (RS) ansatz, one reads

$$
Q_{ls}^{\nu\nu} = Q_{ls}, \ E_{ls}^{\nu\nu} = E_{ls}, \ R_{ls}^{\nu\nu} = R_{ls}, \ m_{sl^*}^\nu = m_{sl^*}, \ b_{sl^*}^\nu = b_{sl^*}, \tag{18}
$$

$$
Q_{ls}^{\nu\kappa} = Q_{ls} - \frac{\chi_{ls}}{\gamma}, \ E_{ls}^{\nu\kappa} = E_{ls} - \frac{\zeta_{ls}}{\gamma}, \ R_{ls}^{\nu\kappa} = R_{ls} - \frac{\omega_{ls}}{\gamma}, \tag{19}
$$

where all parameters are denoted as $\Theta \triangleq (Q, E, R, m, b, \chi, \zeta, \omega) \in \mathbb{R}^{k(6k+2k^*)}$. This RS ansatz restricts the integration of the replicated weight parameters $\{W_\nu, V_\nu\}$ across the entire $\mathbb{R}^{p(2k \times d)}$ to a subspace that satisfies the constraints in Eq. 18 and 19. Using the Fourier transform of the delta functions, $\mathcal{S}$ is expanded as

$$\mathcal{S} = \int d\hat{\Theta} \prod_\nu dW^\nu dV^\nu \, e^{\frac{1}{2}\sum_{ls,\nu}(\gamma\hat{Q}_{ls}-\gamma^2\hat{\chi}_{ls})(dQ_{ls}-\boldsymbol{w}_l^\nu\boldsymbol{w}_s^\nu)-\frac{1}{2}\sum_{ls}\sum_{\nu\neq\kappa}\gamma^2\hat{\chi}\left(Q_{ls}-\frac{\chi_{ls}}{\gamma}-\boldsymbol{w}_l^\nu\boldsymbol{w}_s^\kappa\right)}$$

$$\times e^{\frac{1}{2}\sum_{ls,\nu}(\gamma\hat{E}_{ls}-\gamma^2\hat{\zeta}_{ls})(dE_{ls}-\boldsymbol{v}_l^\nu\boldsymbol{v}_s^\nu)-\frac{1}{2}\sum_{ls}\sum_{\nu\neq\kappa}\gamma^2\hat{\zeta}\left(E_{ls}-\frac{\zeta_{ls}}{\gamma}-\boldsymbol{v}_l^\nu\boldsymbol{v}_s^\kappa\right)}$$

$$\times e^{\sum_{ls,\nu}(\gamma\hat{R}_{ls}-\gamma^2\hat{\omega}_{ls})(dR_{ls}-\boldsymbol{w}_l^\nu\boldsymbol{v}_s^\nu)-\sum_{ls}\sum_{\nu\neq\kappa}\gamma^2\hat{\omega}\left(R_{ls}-\frac{\omega_{ls}}{\gamma}-\boldsymbol{w}_l^\nu\boldsymbol{v}_s^\kappa\right)}$$

$$\times e^{-\sum_{ls}\sum_\nu \gamma\hat{m}_{sl*}(dm_{sl*}-\boldsymbol{w}_s^\nu\boldsymbol{w}_{l*}^*)-\sum_{ls}\sum_\nu\gamma\hat{b}_{sl*}(db_{sl*}-\boldsymbol{v}_s^\nu\boldsymbol{w}_{l*}^*)}e^{-\frac{\gamma\lambda}{2}\sum_\nu\left(\|W_\nu\|_F^2+\|V^\nu\|_F^2\right)}$$

$$= \int d\hat{\Theta}e^{\frac{p\gamma d}{2}\left(\text{tr}\left(\hat{Q}Q+(p-1)\hat{\chi}\chi-p\gamma Q\hat{\chi}\right)+\text{tr}\left(\hat{E}E+(p-1)\hat{\zeta}\zeta-p\gamma E\hat{\zeta}\right)+2\text{tr}\left(\hat{R}R+(p-1)\hat{\omega}\omega-p\gamma R\hat{\omega}\right)-2\text{tr}(\hat{m}^\top m)-2\text{tr}(\hat{b}^\top b)\right)}$$

$$\times \left(\int\prod_\nu d\tilde{\boldsymbol{w}}^\nu e^{-\frac{\gamma}{2}\sum_{ls}\left((\hat{Q}_{ls}+\lambda I_k)\sum_\nu\boldsymbol{w}_l^\nu\boldsymbol{w}_s^\nu+(\hat{E}_{ls}+\lambda I_k)\sum_\nu\boldsymbol{v}_l^\nu\boldsymbol{v}_s^\nu+2\hat{R}_{ls}\sum_\nu\boldsymbol{w}_l^\nu\boldsymbol{v}_s^\nu\right)}\right.$$

$$\times e^{\frac{\gamma^2}{2}\sum_{ls}\left(\hat{\chi}_{ls}\sum_\nu\boldsymbol{w}_s^\nu\sum_\nu\boldsymbol{w}_l^\nu+\hat{\zeta}_{ls}\sum_\nu\boldsymbol{v}_s^\nu\sum_\nu\boldsymbol{v}_l^\nu+2\hat{\omega}_{ls}\sum_\nu\boldsymbol{w}_l^\nu\sum_\nu\boldsymbol{v}_s^\nu\right)}$$

$$\left.\times e^{+\gamma\sum_{l*s}\hat{m}_{sl*}\sum_\nu\boldsymbol{w}_s^\nu\boldsymbol{w}_{l*}^*+\gamma\sum_{l*s}\hat{d}_{sl*}\sum_\nu\boldsymbol{v}_s^\nu\boldsymbol{w}_{l*}^*}\right)^d,$$

where $d\hat{\Theta} \triangleq d\hat{Q}d\hat{E}d\hat{R}d\hat{\chi}d\hat{\zeta}d\hat{m}d\hat{b}$ and $\tilde{\boldsymbol{w}}^\nu \triangleq (w_1^\nu, \ldots, w_k^\nu, v_1^\nu, \ldots, v_k^\nu)$. This can be derived with the help of the identity for any symmetric positive matrix $M \in \mathbb{R}^{k\times k}$ and any vector $\boldsymbol{x} \in \mathbb{R}^k$, given by

$$e^{\frac{1}{2}\boldsymbol{x}^\top M\boldsymbol{x}} = \int D\boldsymbol{\xi}_k e^{\boldsymbol{\xi}_k^\top M^{\frac{1}{2}}\boldsymbol{x}},$$

where $D\boldsymbol{\xi}_{2k}$ is the standard Gaussian measure on $\mathbb{R}^{2k}$. Then, we obtain:

$$\mathcal{S} = \int d\hat{\Theta}e^{\frac{p\gamma d}{2}\left(\text{tr}\left(\hat{Q}Q+(p-1)\hat{\chi}\chi-p\gamma Q\hat{\chi}\right)+\text{tr}\left(\hat{E}E+(p-1)\hat{\zeta}\zeta-p\gamma E\hat{\zeta}\right)+2\text{tr}\left(\hat{R}R+(p-1)\hat{\omega}\omega-p\gamma R\hat{\omega}\right)-\text{tr}(\hat{m}m)-\text{tr}(\hat{b}b)\right)}$$

$$\times \left(\int D\boldsymbol{\xi}_{2k}\left(\int d\tilde{\boldsymbol{w}}e^{-\frac{\gamma}{2}\tilde{\boldsymbol{w}}^\top(\hat{G}+\lambda I_{2k})\tilde{\boldsymbol{w}}+\gamma(\boldsymbol{\xi}_{2k}^\top\hat{g}^{\frac{1}{2}}+\mathbf{1}_{k*}^\top\hat{\phi}^\top)\tilde{\boldsymbol{w}}}\right)^p\right)^d$$

$$= \int d\hat{\Theta}e^{\frac{p\gamma d}{2}\left(\text{tr}\left(\hat{Q}Q+(p-1)\hat{\chi}\chi-p\gamma Q\hat{\chi}\right)+\text{tr}\left(\hat{E}E+(p-1)\hat{\zeta}\zeta-p\gamma E\hat{\zeta}\right)+2\text{tr}\left(\hat{R}R+(n-1)\hat{\omega}\omega-n\gamma R\hat{\omega}\right)-\text{tr}(\hat{m}m)-\text{tr}(\hat{d}d)\right)}$$

$$\times e^{d\log\int D\boldsymbol{\xi}_{2k}\left(\int d\tilde{\boldsymbol{w}}e^{-\frac{\gamma}{2}\tilde{\boldsymbol{w}}^\top(\hat{G}+\lambda I_{2k})\tilde{\boldsymbol{w}}+\gamma(\boldsymbol{\xi}_{2k}^\top\hat{g}^{\frac{1}{2}}+\mathbf{1}_{k*}^\top\hat{\phi}^\top)\tilde{\boldsymbol{w}}}\right)^n}$$

$$= \int d\hat{\Theta}e^{\frac{n\gamma d}{2}\left(\text{tr}\left(\hat{Q}Q-\hat{\chi}\chi\right)+\text{tr}\left(\hat{E}E-\hat{\zeta}\zeta\right)+2\text{tr}\left(\hat{R}R-\hat{\omega}\omega\right)-\text{tr}(\hat{m}m)-\text{tr}(\hat{d}d)+\mathcal{O}(n)\right)}$$

$$\times e^{dn\left(\int D\boldsymbol{\xi}_{2k}\log\int d\tilde{\boldsymbol{w}}e^{-\frac{\gamma}{2}\tilde{\boldsymbol{w}}^\top(\hat{G}+\lambda I_{2k})\tilde{\boldsymbol{w}}+\gamma(\boldsymbol{\xi}_{2k}^\top\hat{g}^{\frac{1}{2}}+\mathbf{1}_{k*}^\top\hat{\phi}^\top)\tilde{\boldsymbol{w}}}+\mathcal{O}(n)\right)}$$

$$= \int d\hat{\Theta}e^{\frac{n\gamma d}{2}\left(\text{tr}\left(\hat{G}G-\hat{g}g\right)-2\text{tr}(\hat{\phi}^\top k)+\text{tr}[(\hat{G}+\lambda I_{2k})^{-1}\hat{g}]+\mathbf{1}_{k*}^\top\hat{\phi}^\top(\hat{G}+\lambda I_{2k})^{-1}\hat{\phi}\mathbf{1}_{k*}\right)+o(n,d,\gamma)}$$

where $\tilde{\boldsymbol{w}} \triangleq (w_1, \ldots, w_k, v_1, \ldots, v_k)$ and

$$\hat{G} \triangleq \begin{pmatrix} \hat{Q} & \hat{R} \\ \hat{R} & \hat{E} \end{pmatrix} \in \mathbb{R}^{2k\times 2k}, \; \hat{g} \triangleq \begin{pmatrix} \hat{\chi} & \hat{\omega} \\ \hat{\omega} & \hat{\zeta} \end{pmatrix} \in \mathbb{R}^{2k\times 2k}, \; \hat{\psi} \triangleq \begin{pmatrix} \hat{m} \\ \hat{b} \end{pmatrix} \in \mathbb{R}^{2k\times k^*}.$$

Note that, under the RS ansatz, $\boldsymbol{h}^\nu$ is expressed as follows

$$\boldsymbol{h}^\nu = \frac{1}{\sqrt{\gamma}}g^{1/2}\boldsymbol{z}^\nu + G^{1/2}\boldsymbol{\xi}, \; \forall\nu\in[p], \; \boldsymbol{z}^\nu\sim\mathcal{N}(\mathbf{0}_{2k}, I_{2k}), \; \boldsymbol{\xi}\sim\mathcal{N}(\mathbf{0}_{2k}, I_{2k}),$$

where

$$G \triangleq \begin{pmatrix} Q & R \\ R & E \end{pmatrix} \in \mathbb{R}^{2k \times 2k}, \ g \triangleq \begin{pmatrix} \xi & \omega \\ \omega & \zeta \end{pmatrix} \in \mathbb{R}^{2k \times 2k}, \ \psi \triangleq \begin{pmatrix} m \\ b \end{pmatrix}.$$

$\mathcal{E}$ is also expanded as

$$\frac{1}{d} \log \mathcal{E} = \frac{1}{d} \log \int \prod_{\nu} dD^{\nu} \left( \int D\boldsymbol{c} \int d\boldsymbol{h} \mathcal{N}(\boldsymbol{h}; \boldsymbol{0}_{2kp}, \Sigma) e^{-\gamma \sum_{\nu} \mathcal{L}(G, g, \psi; \boldsymbol{h}, \boldsymbol{c}, \beta_{\mathrm{VAE}})} \right)^n$$

$$= \frac{p}{d} \log \int dD e^{-\frac{\gamma n}{2}(\mathrm{tr}[(Q + \beta_{\mathrm{VAE}} I_k) D] - \beta_{\mathrm{VAE}} \mathrm{tr}(\log D))}$$

$$+ \alpha \log \mathbb{E}_{\boldsymbol{c}} \int d\boldsymbol{h} \mathcal{N}(\boldsymbol{h}; \boldsymbol{0}_{2pk}, \Sigma) e^{-\gamma \sum_{\nu} \hat{\mathcal{L}}(G, g, \psi; \boldsymbol{h}, \boldsymbol{c}, \beta_{\mathrm{VAE}})}$$

$$= \frac{p}{d} \log \int dD e^{-\frac{\gamma n}{2}(\mathrm{tr}[(Q + \beta_{\mathrm{VAE}} I_k) D] - \beta_{\mathrm{VAE}} \mathrm{tr}(\log D))}$$

$$+ \alpha \log \mathbb{E}_{\boldsymbol{c}, \boldsymbol{\xi}_{2k}} \left( \int D\boldsymbol{z}_{2k} e^{-\gamma \hat{\mathcal{L}}(G, g, \psi; \boldsymbol{z}_{2k}, \boldsymbol{\xi}_{2k}, \boldsymbol{c}, \beta_{\mathrm{VAE}})} \right)^p$$

$$= \frac{p}{d} \log \int dD e^{-\frac{\gamma n}{2}(\mathrm{tr}[(Q + \beta_{\mathrm{VAE}} I_k) D] - \beta_{\mathrm{VAE}} \mathrm{tr}(\log D))}$$

$$+ \alpha p \mathbb{E}_{\boldsymbol{c}, \boldsymbol{\xi}_{2k}} \log \int D\boldsymbol{z}_{2k} e^{-\gamma \hat{\mathcal{L}}(G, g, \psi; \boldsymbol{z}_{2k}, \boldsymbol{\xi}_{2k}, \boldsymbol{c}, \beta_{\mathrm{VAE}})} + o(p),$$

where

$$-\hat{\mathcal{L}}(G, g, \psi; \boldsymbol{z}_{2k}, \boldsymbol{\xi}_{2k}, \boldsymbol{c}, \beta_{\mathrm{VAE}}) = \frac{(\sqrt{\rho} m \boldsymbol{c} + \sqrt{\eta} \boldsymbol{u})^{\top}(\sqrt{\rho} b \boldsymbol{c} + \sqrt{\eta} \tilde{\boldsymbol{u}})}{\sigma^2}$$

$$- \frac{(\sqrt{\rho} b \boldsymbol{c} + \sqrt{\eta} \tilde{\boldsymbol{u}})^{\top}(Q + \sigma^2 \beta_{\mathrm{VAE}} I_k)(\sqrt{\rho} b \boldsymbol{c} + \sqrt{\eta} \tilde{\boldsymbol{u}})}{2\sigma^2}.$$

Then we evaluate the last term as follows:

$$\int D\boldsymbol{c}_k \int D\boldsymbol{\xi}_{2k} \log \int D\boldsymbol{z}_{2k} e^{-\gamma \hat{\mathcal{L}}(G, g, \psi; \boldsymbol{z}_{2k}, \boldsymbol{\xi}_{2k}, \boldsymbol{c}, \beta_{\mathrm{VAE}}, \lambda)}$$

$$= \frac{\gamma \rho}{2\sigma^2} \int D\boldsymbol{c}(\boldsymbol{c}^{\top}(2m^{\top} b - b^{\top}(Q + \sigma^2 \beta_{\mathrm{VAE}} I_k) b) \boldsymbol{c})$$

$$+ \mathbb{E}_{\boldsymbol{c}, \boldsymbol{\xi}_{2k}} \log \int D\boldsymbol{z}_{2k} e^{-\gamma \left( -\frac{1}{2\sigma^2} \left( \frac{g^{1/2} \boldsymbol{z}_{2k}}{\sqrt{\gamma}} + G^{1/2} \boldsymbol{\xi}_{2k} \right)^{\top} A \left( \frac{g^{1/2} \boldsymbol{z}_{2k}}{\sqrt{\gamma}} + G^{1/2} \boldsymbol{\xi}_{2k} \right) + b^{\top} \left( \frac{g^{1/2} \boldsymbol{z}_{2k}}{\sqrt{\gamma}} + G^{1/2} \boldsymbol{\xi}_{2k} \right) \right)}$$

$$= \frac{\gamma \rho}{2\sigma^2} \mathrm{tr} \left( 2m^{\top} b - b^{\top}(Q + \sigma^2 \beta_{\mathrm{VAE}} I_k) d \right) + \frac{\gamma}{\sigma^2} \mathbb{E}_{\boldsymbol{c}, \boldsymbol{\xi}_{2k}} \left( \frac{1}{2} \boldsymbol{\xi}_{2k}^{\top} G^{1/2} A G^{1/2} \boldsymbol{\xi}_{2k} - \boldsymbol{b}^{\top} G^{1/2} \boldsymbol{\xi}_{2k} \right)$$

$$+ \mathbb{E}_{\boldsymbol{c}, \boldsymbol{\xi}_{2k}} \log \int d\boldsymbol{z}_{2k} e^{\gamma \left( -\frac{1}{2} \boldsymbol{z}_{2k}^{\top}(I_{2k} - g^{1/2} A g^{1/2}) \boldsymbol{z}_{2k} + (\boldsymbol{\xi}_{2k}^{\top} G^{1/2} A - \boldsymbol{b}^{\top}) g^{1/2} \boldsymbol{z}_{2k} \right)}$$

$$= \frac{\gamma \rho}{2\sigma^2} \mathrm{tr} \left( 2b^{\top} m - b^{\top}(Q + \sigma^2 \beta_{\mathrm{VAE}} I_k) b \right) + \frac{\gamma}{2\sigma^2} \mathrm{tr}(AG)$$

$$+ \frac{\gamma}{2\sigma^2} \mathbb{E}_{\boldsymbol{c}, \boldsymbol{\xi}_{2k}} (\boldsymbol{\xi}_{2k}^{\top} G^{1/2} A - \boldsymbol{b}^{\top}) g^{1/2} (I_{2k} - g^{1/2} A g^{1/2})^{-1} g^{1/2} (A G^{1/2} \boldsymbol{\xi}_{2k} - \boldsymbol{b}) + o(\gamma)$$

$$= \frac{\gamma}{2\sigma^2} \left( \rho \mathrm{tr} \left( 2b^{\top} m - b^{\top}(Q + \sigma^2 \beta_{\mathrm{VAE}}) b \right) + \mathrm{tr}(AG) + \mathrm{tr} \left( (I_{2k} - Ag)^{-1}(AGA + BB^{\top}) g \right) \right),$$

where

$$A = \eta \begin{pmatrix} \boldsymbol{0}_{k \times k} & I_k \\ I_k & -(Q + \sigma^2 \beta_{\mathrm{VAE}} I_k) \end{pmatrix}, \quad \boldsymbol{b} = B\boldsymbol{c}, \quad B = \sqrt{\rho \eta} \begin{pmatrix} -b \\ -m + (Q + \sigma^2 \beta_{\mathrm{VAE}} I_k) b \end{pmatrix}$$

Taking the limit $\gamma \to \infty$, one can obtain

$$\log \mathcal{E} = \frac{dp\gamma\alpha}{2\sigma^2}\left(\rho\mathrm{tr}\left(2b^\top m - b^\top(Q + \sigma^2\beta_{\mathrm{VAE}})b\right) + \mathrm{tr}(AG) + \mathrm{tr}\left((I_{2k} - Ag)^{-1}(AGA + BB^\top)g\right)\right.$$

$$\left. + \sigma^2 \sum_k \log \frac{e(Q_{kk} + \beta_{\mathrm{VAE}})}{\beta_{\mathrm{VAE}}}\right)$$

Substituting $\mathcal{S}$ and $\mathcal{E}$ into the expression of the replicated partition function yields

$$\mathbb{E}_{\mathcal{D}}Z^p(\mathcal{D},\gamma) = \int d\Theta d\hat{\Theta}e^{\frac{p\gamma d}{2}\left(\mathrm{tr}(\hat{G}G - \hat{g}g) - 2\mathrm{tr}(\hat{\phi}^\top k) + \mathrm{tr}[(\hat{G} + \lambda I_{2k})^{-1}\hat{g}] + \mathbf{1}_{k*}^\top\hat{\phi}^\top(\hat{G} + \lambda I_{2k})^{-1}\hat{\phi}\mathbf{1}_{k*}\right)}$$

$$\times e^{\frac{dp\gamma\alpha}{2\sigma^2}\left(\rho\mathrm{tr}\left(2b^\top m - b^\top(Q + \beta_{\mathrm{VAE}})b\right) + \mathrm{tr}(AG) + \mathrm{tr}\left((I_{2k} - Ag)^{-1}(AGA + BB^\top)g\right) + \sigma^2\sum_k \log \frac{e(Q_{kk} + \beta_{\mathrm{VAE}})}{\beta_{\mathrm{VAE}}}\right)}$$

In the end, from the identity:

$$\lim_{p \to +0} \frac{\log \mathbb{E}_{\mathcal{D}}Z(\mathcal{D},\gamma)^p}{p},$$

one obtains

$$f = \frac{1}{2}\underset{\substack{G,g,\psi\\\hat{G},\hat{g},\hat{\psi}}}{\mathrm{extr}}\left\{\mathrm{tr}\left[g\hat{g} + 2\psi\hat{\psi} - G\hat{G}\right] - \mathrm{tr}\left[(\hat{G} + \lambda)^{-1}\hat{g}\right] - \mathbf{1}_{k*}^\top\hat{\psi}^\top(\hat{G} + \lambda)^{-1}\hat{\psi}\mathbf{1}_{k*}\right.$$

$$\left. + \alpha\left(\mathrm{tr}\left[AG - \sqrt{\frac{\rho}{\eta}}\psi^\top B + (I_{2k} - Ag)^{-1}(AGA + BB^\top)g\right] + \sum_{l=1}^{k}\log \frac{e(Q_{ll} + \beta_{\mathrm{VAE}})}{\beta_{\mathrm{VAE}}}\right)\right\}$$

(20)

where $\mathrm{extr}$ indicates taking the extremum with respect to $\Theta$. This concludes the whole proof of Eq. (12).

## D.3 FREE-ENERGY DENSITY $k = k^* = 1$

When $k = k^* = 1$, a part of the exponential function of Eq. (12) can be reduced as

$$-\frac{1}{2}\left(\mathrm{tr}[(\hat{G} + \lambda)^{-1}\hat{g}] + \mathbf{1}_{k*}^\top\hat{\psi}^\top(\hat{G} + \lambda)^{-1}\hat{\psi}\mathbf{1}_{k*}\right)$$

$$= -\frac{(\lambda + \hat{E})(\hat{m}^2 + \hat{\chi}) + (\lambda + \hat{Q})(\hat{b}^2 + \hat{\zeta}) - 2\hat{R}(\hat{m}\hat{b} + \hat{\omega})}{2((\hat{Q} + \lambda)(\hat{E} + \lambda) - \hat{R}^2)}. \quad (21)$$

Next, we evaluate the energy term. Initially, when $k = k^* = 1$, the following expression holds:

$$G^{\frac{1}{2}} = \frac{1}{\sqrt{Q + E + 2\sqrt{QE - R^2}}}\begin{pmatrix} Q + \sqrt{QE - R^2} & R \\ R & E + \sqrt{QE - R^2} \end{pmatrix},$$

$$(I_{2k} + Ag)^{-1} = \frac{1}{\eta\zeta(Q - \eta\chi + \beta_{\mathrm{VAE}}) + (\eta\omega - 1)^2}\begin{pmatrix} \eta\zeta(Q + \beta_{\mathrm{VAE}}) + 1 - \eta\omega & \eta\zeta \\ \eta(\chi - (Q + \beta_{\mathrm{VAE}})\omega) & 1 - \eta\omega \end{pmatrix}.$$

By substituting these into the formula for energy term in Eq. (12), the following free energy can be derived:

$$f = \underset{\Theta}{\mathrm{extr}}\left\{-\frac{1}{2}(\hat{G}G - g\hat{g}) + \hat{\psi}^\top\psi + \frac{(\lambda + \hat{E})(\hat{m}^2 + \hat{\chi}) + (\lambda + \hat{Q})(\hat{b}^2 + \hat{\zeta}) - 2\hat{R}(\hat{m}\hat{m} + \hat{\omega})}{2\hat{G}}\right.$$

$$-\frac{\alpha}{2}\left(\frac{(Q - \eta\chi + \beta_{\mathrm{VAE}})(\rho b^2 + \eta E) - \eta\zeta(\rho m^2 + \eta Q)}{G}\right.$$

$$\left.\left. + \frac{2(\eta\omega - 1)(\rho mb + \eta r)}{G} + \beta_{\mathrm{VAE}}\log \frac{e(Q + \beta_{\mathrm{VAE}})}{\beta_{\mathrm{VAE}}}\right)\right\}. \quad (22)$$

From the free-energy gradient, the extremum conditions are explicitly given by

$$Q = \frac{(\hat{E} + \lambda)\hat{H}}{\hat{G}^2} - \frac{\hat{b}^2 + \hat{\zeta}}{\hat{G}}, \; E = \frac{(\hat{Q} + \lambda)\hat{H}}{\hat{G}^2} - \frac{\hat{m}^2 + \hat{\chi}}{\hat{G}},$$

$$R = -\frac{\hat{R}\hat{H}}{\hat{G}^2} + \frac{\hat{m}\hat{b} + \hat{\omega}}{\hat{G}},$$

$$m = \frac{\hat{m}(\hat{E} + \lambda) - \hat{b}\hat{R}}{\hat{G}}, \; \tilde{m} = \frac{\hat{m}(\hat{E} + \lambda) - \hat{m}\hat{R}}{\hat{G}},$$

$$\chi = \frac{\hat{E} + \lambda}{\hat{G}}, \; \tilde{\chi} = \frac{\hat{Q} + \lambda}{\hat{G}}, \; \omega = -\frac{\hat{R}}{\hat{G}},$$

$$\hat{Q} = \alpha \left( \frac{\beta_{\text{VAE}}}{Q + \beta_{\text{VAE}}} + \frac{\eta Q + b^2 \rho - \eta^2 \chi}{G} - \frac{\eta \zeta H}{G^2} \right),$$

$$\hat{E} = \alpha\eta \left( \frac{Q - \eta\chi + \beta_{\text{VAE}}}{G} \right), \; \hat{R} = \alpha\eta \left( \frac{\eta\omega - 1}{G} \right),$$

$$\hat{\chi} = \alpha\eta \left( \frac{G(\eta E + b^2 \rho) - \eta \zeta H}{G^2} \right),$$

$$\hat{\zeta} = \alpha\eta \left( \frac{G(\eta Q + m^2 \rho) - \eta\chi H}{G^2} \right),$$

$$\hat{\omega} = \alpha\eta \left( \frac{-G(\eta R + mb\rho) + (\eta\omega - 1)H}{G^2} \right),$$

$$\hat{m} = \alpha\rho \left( \frac{\eta m\chi - b(\eta\omega - 1)}{G} \right),$$

$$\hat{d} = -\alpha\rho \left( \frac{d(Q - \eta\chi + \beta_{\text{VAE}}) + m(\eta\omega - 1)}{G} \right),$$

where

$$\hat{G} = (\hat{Q} + \lambda)(\hat{E} + \lambda) - \hat{R}^2$$
$$G = \eta\zeta(Q - \eta\chi + \beta_{\text{VAE}}) + (\eta\omega - 1)^2$$
$$\hat{H} = (\lambda + \hat{E})(\hat{m}^2 + \lambda) + (\lambda + \hat{Q})(\hat{d}^2 + \hat{\zeta}) - 2\hat{R}(\hat{m}\hat{d} + \hat{\omega}),$$
$$H = (d^2\rho + \eta E)(Q - \eta\chi + \beta_{\text{VAE}}) - \eta\zeta(m^2\rho + \eta Q) + 2(\eta R + md\rho)(\rho\omega - 1).$$

Thus, the signal recovery error and other summary statistics can be evaluated by numerically solving the self-consistent equations.

### D.4 DERIVATION OF CLAIM 6.1

**Case:** $k = k^* = 1$. From the expansion in the first order term with respect to $\alpha$, one obtains the following solution from Eq. (12):

$$Q = E = R = \chi = \zeta = \omega = m = b = 0 \quad (\rho + \eta \leq \beta_{\text{VAE}}), \tag{23}$$

$$Q = \eta + \rho - \beta_{\text{VAE}}, \; E = \frac{\eta + \rho - \beta_{\text{VAE}}}{(\eta + \rho)^2}, \; \chi = \zeta = \omega = 0, \tag{24}$$

$$m = \sqrt{\eta + \rho - \beta_{\text{VAE}}}, \; b = \frac{\eta + \rho - \beta_{\text{VAE}}}{\eta + \rho} \quad (\rho + \eta > \beta_{\text{VAE}}). \tag{25}$$

Note that one can evaluate $\lim_{\gamma \to \infty} \mathbb{E}_{\mathcal{D}} \mathbb{E}_{p(W,V,D;\mathcal{D},\gamma)} \varepsilon_g(W, W^*)$ as $\rho - 2\sqrt{\rho}m + Q$. Thus, one obtains

$$\epsilon_g = \begin{cases} \rho - \sqrt{\eta + \rho - \beta_{\text{VAE}}}(2\sqrt{\rho} - \sqrt{\eta + \rho - \beta_{\text{VAE}}}) & (\rho + \eta \leq \beta_{\text{VAE}}) \\ \rho & (\rho + \eta > \beta_{\text{VAE}}) \end{cases}. \tag{26}$$

The optimal condition for $\beta_{\text{VAE}}$ yields optimal value $\beta^*_{\text{VAE}} = \eta$.

**Case: General $k = k^*$.** We next prove the generalization of the case $k = k^* > 1$. The saddle-point equations from Eq. (12) are expanded in the limit $\alpha \to \infty$, yielding the following relationships:

$$(\psi)_{ls} = \mathcal{O}(\alpha^0), \quad \forall l, s \in [k],$$

$$(G)_{ls} = \mathcal{O}(\alpha^0), \quad \forall l, s \in [k],$$

$$(g)_{ls} = \mathcal{O}(\alpha^{-1}), \quad \forall l, s \in [k],$$

$$(\hat{g})_{ls} = \mathcal{O}(\alpha^{-1}), \quad \forall l, s \in [k].$$

From these equations, we find that $g = \mathbf{0}_{k \times k}$ and $\hat{g} = \mathbf{0}_{k \times k}$. Moreover, in this limit, the contribution from regularization becomes negligible. Therefore, by setting $\lambda = 0$, the free-energy density can be expressed as follows:

$$f = \frac{1}{2} \underset{G, \psi, \hat{G}, \hat{\psi}}{\text{extr}} \left\{ \text{tr}(G\hat{G}) - 2\text{tr}(\psi\hat{\psi}) + \mathbf{1}_k^\top \hat{\psi}^\top \hat{G}^{-1} \hat{\psi} \mathbf{1}_k \right.$$

$$\left. + \alpha \left( \text{tr}\left[ AG + \rho \left( 2b^\top m - b^\top (Q + \beta_{\text{VAE}})b \right) \right] + \sum_l \log \frac{e(Q_{ll} + \beta_{\text{VAE}})}{\beta_{\text{VAE}}} \right) \right\}.$$

From the saddle-point equations, the following relations are derived:

$$\psi = \hat{G}^{-1} \hat{\psi} \mathbf{1}_k \mathbf{1}_k^\top, \quad G = \hat{G}^{-1} \hat{\psi} \mathbf{1}_k \mathbf{1}_k^\top \hat{\psi}^\top \hat{G}^{-1}, \quad \hat{G} = -\alpha A,$$

From the relations, we find $G = \psi\psi^\top$. Using these relations, the free-energy density can be represented as an extremum with respect to $m$ and $b$:

$$f = \frac{1}{2} \underset{m, b, \hat{m}, \hat{b}}{\text{extr}} \left\{ -2\text{tr}(m\hat{m}^\top + b\hat{b}) - \frac{1}{\alpha\eta} \mathbf{1}_k^\top \left( \hat{m}^\top (mm^\top + \beta_{\text{VAE}} I_k)\hat{m} + 2\hat{b}^\top \hat{m} \right) \mathbf{1}_k \right.$$

$$\left. + \alpha \left( \text{tr}\left[ \rho(2b^\top m - b^\top(mm^\top + \beta_{\text{VAE}})b) \right] + \sum_l \log \frac{e(m_{ll}^2 + \beta_{\text{VAE}})}{\beta_{\text{VAE}}} \right) \right\}.$$

From the saddle-point condition, the following relations are derived:

$$\hat{m} = -\frac{1}{\alpha\eta} mm^\top \hat{m} \mathbf{1}_k \mathbf{1}_k^\top + \alpha\rho b(b^\top m - 1) + \alpha\text{diag}\left( \left\{ \frac{m_{ll}}{m_{ll}^2 + \beta_{\text{VAE}}} \right\} \right),$$

$$\hat{b} = \alpha\rho((mm^\top + \beta_{\text{VAE}} I_k)b - m),$$

$$m = -\frac{1}{\alpha\eta} \left( mm^\top + \beta_{\text{VAE}} I_k \right) \hat{m} \mathbf{1}_k \mathbf{1}_k^\top,$$

$$b = -\frac{1}{\alpha\eta} \hat{m} \mathbf{1}_k \mathbf{1}_k^\top.$$

Considering the fact that in the data generation process, $W^* = I_{k^*}$ and $n$ follows a standard Gaussian distribution, it is reasonable to assume that $W$ and $V$ become diagonal matrices after learning as $\alpha \to \infty$, i.e., the off-diagonal elements of $Q$ and $E$ become zero. Under this assumption, the following can be derived from the saddle-point equations:

$$m_l = \frac{m_l \rho}{\beta_{\text{VAE}} + (m_l^2 b_l^2 - 1)\eta + b_l^2(m_l^2 \rho + \beta_{\text{VAE}}(\eta + \rho))}, \quad \forall l \in [k]$$

$$b_l = \frac{\eta\rho b_l + (m_l - b(m_l^2 + \beta_{\text{VAE}}))\rho \left( \frac{\beta_{\text{VAE}}}{m_l^2 + \beta_{\text{VAE}}} + b_l^2(\eta + \rho) \right)}{\eta(\beta_{\text{VAE}} - \eta + b_l^2(m_l^2 + \beta_{\text{VAE}})(\eta + \rho))}, \quad \forall l \in [k].$$

This system of equations admits both the posterior-collapse solution $\boldsymbol{m} = \mathbf{0}_k$, $\boldsymbol{b} = \mathbf{0}_k$ and the Learnable solution $\boldsymbol{m} = \sqrt{\rho + \eta - \beta_{\text{VAE}}} \mathbf{1}_k$, $\boldsymbol{b} = \sqrt{\rho + \eta - \beta_{\text{VAE}}}/\rho + \eta \mathbf{1}_k$. Since these equations are decoupled for each $l$, we focus below on analyzing the linear stability of the posterior-collapse solution for a specific $l$. Linearizing around the posterior-collapse solution, we obtain the following:

$$\begin{pmatrix} \delta m_l \\ \delta b_l \end{pmatrix} = \frac{\rho}{\beta_{\text{VAE}} - \eta} \begin{pmatrix} 1 & 0 \\ 1/\eta & 1 \end{pmatrix} \begin{pmatrix} \delta m_l \\ \delta b_l \end{pmatrix}.$$

The condition where the Jacobian eigenvalue becomes 1 corresponds to the destabilized region. The threshold, as in the case of $k = k^* = 1$, is given by $\beta_{\text{VAE}} = \rho + \eta$.

## D.5 DERIVATION OF CLAIM 6.2

We first notice that rate and distortion can be expressed as

$$R = \mathbb{E}_{\mathcal{D}} R(\hat{W}(\mathcal{D}), \hat{V}(\mathcal{D}), \hat{D}(\mathcal{D})) = \frac{1}{2}\left(\rho b^2 + \eta E + \frac{\beta_{\text{VAE}}}{Q + \beta_{\text{VAE}}} - 1 - \log\frac{\beta_{\text{VAE}}}{Q + \beta_{\text{VAE}}}\right), \quad (27)$$

$$D = \mathbb{E}_{\mathcal{D}} D(\hat{W}(\mathcal{D}), \hat{V}(\mathcal{D}), \hat{D}(\mathcal{D})) = \frac{1}{2}\left(\rho + \eta - 2(\rho m b + \eta R) + Q\left((\rho b^2 + \eta E) + \frac{\beta_{\text{VAE}}}{Q + \beta_{\text{VAE}}}\right)\right), \quad (28)$$

respectively. Then, substituting Eq. (23) and (23) into Eq. (27) and Eq. (28), one can obtain

$$R = \begin{cases} \frac{1}{2}\log\frac{\eta+\rho}{\beta_{\text{VAE}}} & \rho + \eta \le \beta_{\text{VAE}} \\ 0 & \rho + \eta > \beta_{\text{VAE}} \end{cases},$$

$$D = \begin{cases} \frac{\beta_{\text{VAE}}}{2} & \rho + \eta \le \beta_{\text{VAE}} \\ \frac{\rho+\eta}{2} & \rho + \eta > \beta_{\text{VAE}} \end{cases}.$$

From these equations, one obtains

$$R(D) = \begin{cases} \frac{1}{2}\log\frac{\rho+\eta}{2D} & 0 \le D < \frac{\eta+\rho}{2} \\ 0 & D \ge \frac{\rho+\eta}{2} \end{cases}.$$

## E EXPERIMENT DETAILS

### E.1 DETAILS ON REAL DATA AND VAES SIMULATIONS

This section provides detailed information on the experiment with real dataset and non-linear VAEs as shown in Fig. 5. All experiments were conducted using a Xeon(R) Gold 6248 CPU with 26 threads and a Tesla T4 GPU.

**Preprocessing** The MNIST (Deng, 2012) and Fashion MNIST (Xiao et al., 2017) dataset were preprocessed by flattening the images into vectors and normalizing the pixel values by dividing each value by 255 rescaled pixel.

**Architecture** For the MNIST and Fashion MNIST, we employed a multi-layer perceptron variational autoencoder (MLPVAE) implemented in `Pytorch`. The MLPVAE was designed to handle input data of dimension 784, corresponding to $28 \times 28$ pixel images flattened into a single vector. The encoder architecture comprised a linear transformation, `Linear(784, 128)`, followed by a ReLU activation function, and then two linear layers, `Linear(128, 2)`, which output the mean $\mu(z) \in \mathbb{R}^2$ and logarithm of the variance $\log\sigma^2(z) \in \mathbb{R}^2$ of the latent space. The decoder reconstructs the input by performing a linear transformation, `Linear(2, 128)`, followed by a ReLU activation function and a final linear layer, `Linear(128, 784)`, to generate the output.

**Training** The MLPVAE model was trained using the mini-batch Adam optimizer (Kingma & Welling, 2013), with a learning rate of 0.001, a weight decay of 0.0001, and a mini-batch size of 256. The model was then trained for 30 epochs.

**FID estimation** To quantitatively evaluate the quality of images generated by the MLPVAE model on the MNIST and Fashion MNIST datasets, we employed the Fréchet Inception Distance (FID) (Heusel et al., 2017). The FID score is a well-established metric for assessing the similarity between two sets of images, measuring the quality of generated images relative to real ones. It achieves this by comparing the distributions of features extracted from an Inception v3 model (Szegedy et al., 2015) for both real and generated images, with lower FID scores indicating higher similarity and better image quality. For the FID calculation, we utilized `torchmetrics.image.fid`, which provides an implementation of the FID computation.

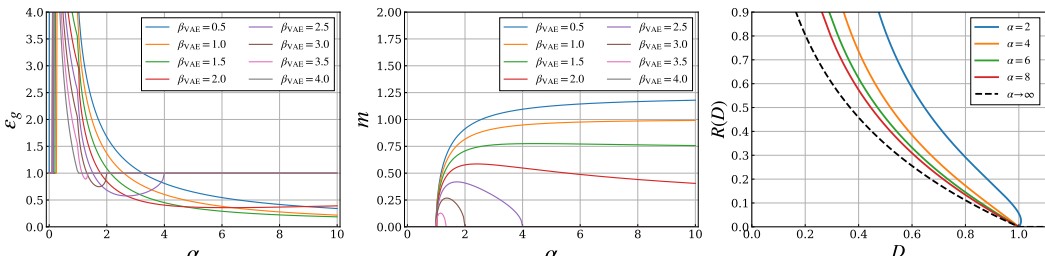

Figure 6: (Left) signal recovery error as a function of sample complexity $\alpha$ for fixed $\lambda = 0$ and varying $\lambda$. (Middle) The summary statistics $m$ with fixed $\lambda = 0$ for different $\beta_{\text{VAE}}$. (Right) RD curve for $\lambda = 0$ with various values of $\alpha$. The dashed line represents the curve in the limit of infinite $\alpha$.

We preprocessed images from both MNIST and FashionMNIST datasets to align with the input requirements for FID calculation. This preprocessing included resizing the images to $299 \times 299$ pixels and converting them to three-channel RGB format. Since MNIST and Fashion MNIST images are originally in grayscale, we converted them to RGB by replicating the single grayscale channel three times. Additionally, we normalized the images using the mean and standard deviation values typically employed for pre-trained models. The FID calculation involved two primary steps. First, we preprocessed both the real and generated images. The real images were sourced directly from the dataset, while the generated images were produced by the trained MLPVAE model. We used $750$ samples each from the real and generated images to estimate the FID score. This sample size was determined to be sufficient for obtaining a reliable estimate, ensuring robust and meaningful comparisons between the real and generated image distributions. Second, we computed the FID score using these preprocessed images. We set the feature parameter to $64$ in the `FrechetInceptionDistance`. This parameter defines the number of features to extract from the images using the Inception network, with $64$ features providing a sufficient representation for accurate FID calculation while balancing computational efficiency.

**Noise strength estimation**  In our theoretical analysis, we assume SCM, i.e., described by probabilistic PCA (Tipping & Bishop, 1999) for the data model, which forms the basis for our estimation of the noise strength $\eta$. Given this assumption, we employs PCA to estimate $\hat{\eta}$, which represents the average variance of the reconstructed data after dimensionality reduction. For both the MNIST and Fashion MNIST datasets, we follow a consistent procedure. We start flatten and normalize them. Applying PCA to these preprocessed images allows us to identify the principal components that capture the majority of the variance in the data. By examining the cumulative variance ratio, we determine the number of principal components required to account for $80\%$ of the total variance then transform the data into the rest of $20\%$, *bulk* and reconstruct them. $\hat{\eta}$ are estimated by the empirical standard deviation of this reconstructed data in bulk.

# F  ADDITIONAL RESULTS

## F.1  EVALUATION OF SIGNAL RECOVERY ERROR AND RD CURVE IN VAE WITHOUT WEIGHT DECAY

This section investigates the signal recovery error and RD curve in the VAE without weight decay when $\rho = \eta = 1$. Fig. 6 (Left) demonstrate the dataset-size dependence of the signal recovery error $\varepsilon_g$ under different $\beta_{\text{VAE}}$ with $\lambda = 0$. Fig. 6 (Middle) shows the dataset-size dependence of the summary statistics $m$ under varied $\beta_{\text{VAE}}$ with $\lambda = 0$. Similar to the results with $\lambda = 1$ in Sec. 6.1, these results indicate that a peak emerges at $\alpha = 1$, and the summary statistics $m$ gradually decreases when $\beta_{\text{VAE}} \geq 2$, leading to posterior collapse. It is important to note that posterior collapse occurs in VAEs even at $\lambda = 0$ when $\beta_{\text{VAE}} = \rho + \eta$, as $\alpha$ approaches infinity because Claim 6.1 consistently holds for any $\lambda$. Subsequently, Fig. 6 (Middle) demonstrates that the RD curve both for the large $\alpha$ limit and for finite $\alpha$ at $\lambda = 0$. As observed with the RD curve at $\lambda = 1$ in Sec. 6.4, achieving the optimal RD curve in regions of smaller distortion necessitates a large dataset.

## F.2 REPLICA PREDICTION AGAINST CIFAR10 AND CONVOLUTIONAL NEURAL NETWORKS

In this section, in addition to the experiments in Section 6.5, we present numerical results using a more realistic setting with CIFAR10 color images (Krizhevsky, 2009) and convolutional neural networks (CNNs). The evaluation methods for FID follow the procedures outlined in Section E.1.

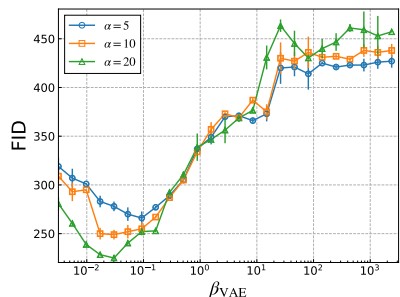

Figure 7: FIDs as a function of $\beta_{\text{VAE}}$ for the CIFAR10 dataset and the CNN. The error bars represent the standard deviations of the results.

CIFAR10 images were kept as 3-channel images due to the use of convolutional neural networks. Rescaling was performed in the same way as with MNIST and FashionMNIST. We implemented a convolutional VAE using `Pytorch`, specifically designed to handle images with three channels. The encoder architecture starts with a series of convolutional layers: `Conv2d(3, 32, kernel_size=4, stride=2, padding=1)` and `Conv2d(32, 64, kernel_size=4, stride=2, padding=1)`, each followed by a ReLU activation function. The output is then flattened into a vector, which is further processed by two linear layers, `Linear(4096, 128)`, that produce the 128-dimensional mean $\mu(z)$ and the 128-dimensional logarithm of the variance $\log \sigma^2(z)$ of the latent space. The decoder reconstructs the input by performing a linear transformation `Linear(128, 4096)`, then reshaping the result into a 3D tensor. This is followed by a series of transposed convolutional layers: `ConvTranspose2d(64, 32, kernel_size=4, stride=2, padding=1)` and `ConvTranspose2d(32, 3, kernel_size=4, stride=2, padding=1)` to generate the output.

Figure 7 presents the FID scores as a function of $\beta_{\text{VAE}}$ under various sample complexities $\alpha = 5, 10$, and 20. The errors represent the standard deviation across three seeds. These results suggest that, as in the results obtained by the replica analysis, the optimal $\beta_{\text{VAE}}$ shifts toward smaller values as the training data increases. Moreover, over around $\beta_{\text{VAE}} \approx 2.62 \times 10^1$, posterior collapse is observed, with no change in performance for larger $\beta_{\text{VAE}}$ values. This observation supports the robustness of our theoretical results, even for complex architectures like CNN-based VAEs.

