# OpenReview forum: "High-dimensional Asymptotics of VAEs: Threshold of Posterior Collapse and Dataset-Size Dependence of Rate-Distortion Curve"
_ICLR.cc/2025/Conference — Submitted to ICLR 2025_

### Official Review · Reviewer_1vWD · 2024-11-01

**Soundness:** 2
**Presentation:** 3
**Contribution:** 2
**Rating:** 3
**Confidence:** 5

**Summary:**

This paper explores the influence of the β hyperparameter in a β-VAE, particularly regarding its impact on the posterior collapse phenomenon and the Rate-Distortion (RD) curve, from an information-theoretic perspective. Using a Spiked Covariance Model (SCM), the authors identify three phases (regularized, learning, overlearning) based on β and sample complexity α = n/d . The results suggest that high values of β inevitably lead to posterior collapse, regardless of data volume, and propose that the optimal β correlates with noise
strength ˆη in high-complexity settings. Experiments on MNIST and Fashion-MNIST datasets partially validate these claims.

**Strengths:**

• Information-Theoretic Perspective: A thoughtful analysis of β-VAE’s Rate-Distortion properties.
• Insight on Hyperparameter Tuning: Suggests an optimal β based on sample complexity, a novel angle.
• Focus on an Underexplored Hyperparameter: Thorough analysis of β’s role in posterior collapse.

**Weaknesses:**

• Over-Simplified Model and Limited Scope: Most analysis centers on a linear VAE with low latent dimensions (e.g., dimension 1), which reduces the generalizability to real-world settings. Minimal insight into non-asymptotic or intermediate regimes.
• Limited Novelty: The core claim that “high β is detrimental” is already known. While it validates an intuition, the work lacks a new method or paradigm. Moreover, it is unclear that the posterior collapse phenomenon occurs ”often” as stated in the abstract, especially when the latent dimension increases.
• Minimal Experiments: Only FID scores on two datasets (MNIST, Fashion-MNIST), without broader validation on diverse or complex data,
and lacks sensitivity analysis (for small changes on the value).
• Identifiability of Ground Truth Model: As is, the ground truth model does not seem to be identifiable (it is when dividing by √θ). This hinders the reliability of the analysis.

**Questions:**

1. Could you explore the performance of this model in transitional or non-asymptotic regimes to reflect real-world data scenarios?
2. Is your analysis still valid for datasets with complex spectrums?
3. Could you clarify the derivation in Equation (9), especially the expectation notation?

---

> ### Author Response · Authors · 2024-11-21
> **Response (1)**
>
> We sincerely thank you for your constructive feedback.
> Below, we address each of the weaknesses and questions you raised to clarify our contributions and propose improvements to the manuscript.
>
> ## **Over-Simplified Model and Limited Scope**
>
> We appreciate your concern regarding the simplicity of our model and its scope.
> As discussed in the RELATED WORK section, Linear VAE is a well-established and effective model for gaining insight into more complex VAEs.
> Indeed, **[1] provided theoretical insights into Linear VAEs, which have led to the development of algorithms for deep models.**
> However, despite their foundational importance,  the dataset-size dependence of the RD curve and generalization performance in Linear VAEs has not been addressed before.
> **Our primary contribution is the analysis of sample complexity dependence ($\alpha$) in Linear VAEs, as noted by Reviewer NKDk, *"this is the first paper that studied RD curves in VAEs as a function of dataset size and data dimensions."**
> Our findings are not only theoretically novel but also practically relevant.
> CONCLUSION section (Line 516-529) in our revised manuscript emphasizes guidance for $\beta_{\mathrm{VAE}}$-tuning derived from our analysis. This robustness is supported by experiments in Section 6.5 on MNIST, Fashion-MNIST, and CIFAR-10 (Appendix F), where the predicted behavior of $\beta_{\mathrm{VAE}}$ qualitatively aligns with real-world data.
>
> ## **Regarding the simple setting $(k=1)$**
>
> Our primary contribution is the derivation of a general formula (Claim 5.2), enabling a sharp analysis of the dataset-size dependence of various metrics in linear VAEs- such as rate $R$, distortion $D$, signal recovery error $\varepsilon_{g}$, $D_{\mathrm{KL}}[p(x) \| p_{\mathrm{data}}(x)]$.
> While we focus on the simplest case with $k=k^{\ast}=1$, this minimal case still captures critical phenomena such as **inevitable posterior collapse** and the **double-descent behavior**. Similar minimal settings with $k=1$ have been the focus of analyses on denoising autoencoders [2].
> While extending the analysis to $k>1$ from the general formula (Claim 5.2) would yield deeper insights into disentanglement, we leave this exploration as future work.
>
> - [1] Juhan Bae et al., Multi-Rate VAE: Train Once, Get the Full Rate-Distortion Curve, ICLR2023
> - [2] Hugo Cui and Lenka Zdeborova, High-dimensional Asymptotics of Denoising Autoencoders, NeurIPS2023
>
> ## **Minimal Insight into Non-Asymptotic or Intermediate Regimes**
>
> Thank you for this insightful observation. Investigating the relevance of asymptotic behavior in intermediate regimes is indeed important.
> To address this, **we validated our theoretical predictions using numerical experiments on a Linear VAE with $d=5{,}000$, showing strong agreement with our asymptotic analysis (Figure 2)**. Additionally, qualitative consistency with real-world datasets such as MNIST, Fashion MNIST, and CIFAR10 further supports the robustness of our findings.
> While our results suggest applicability to non-asymptotic regimes, the extent of this applicability—particularly for smaller $d$—is an interesting question for future work.
>
> ## **Limited Novelty**
>
> We agree that posterior collapse for high $\beta_{\mathrm{VAE}}$ might not be surprising.
> **However, as Reviewer Xf1j noted, "Figure 2 also shows a long plateau in the reconstruction error for large values of $\beta$ . This is backed by Claim 6.1 (in the large $\beta$ limit) and lines 428-430 provide concrete guidance to practitioners about the risks of a large $\beta$ when training."**
> Additionally, identifying a double-descent phenomenon in VAEs, akin to supervised learning, represents a novel theoretical contribution that enhances our understanding of learning dynamics. Indeed, **As Reviewer NKDk highlighted, "This topic is a valuable topic of study and will indeed be of interest to the ICLR community."**
>
> In response to your suggestion, we have **revised CONCLUSION section (Line 516-529)** to explicitly summarize the practical implications of our findings, focusing on the practical tuning of $\beta_{\mathrm{VAE}}$ for engineering applications of VAEs. This revision connects our theoretical insights to their real-world relevance, bridging the gap between abstract findings and practical usability.

---

> ### Author Response · Authors · 2024-11-21
> **Response (2)**
>
> ## **Minimal Experiments**
>
> The primary contribution of our paper is the derivation of a general formula, as presented in Claim 4.2, to analyze the sample complexity dependence of VAEs. Previous studies, mentioned in the main text (Line 495-497 in Section 6.5) and [7], that examine sample complexity $\alpha$ dependence typically focus only on MNIST and CIFAR10 datasets and support the Gaussian universality [3] for deterministic autoencoders, which suggests that real data phenomena can be explained by the data generation process of a Gaussian model. **Accordingly, we employed this setting and consideration for our numerical experiments.**
> While it is essential for the field to verify theoretical consistency across more comprehensive datasets, our main contribution lies in the theoretical analysis. Although our numerical experiments are limited, a more exhaustive empirical validation is part of future work.
>
> - [3] Maria Refinetti and Sebastian Goldt, The dynamics of representation learning in shallow, non-linear autoencoders, ICML2022
>
> ## **Identifiability of Ground Truth Model**
>
> We would appreciate further clarification regarding the definition of identifiability in this context, along with relevant references and an explanation of why it may pose a problem in our setting. The spiked covariance model we employ is widely used in theoretical studies of unsupervised learning discussed in RELATED WORK (High-dimensional asymptotics from replica method ), and denoising autoencoders [2] , autoencoders [3]. Its validity and relevance have been established in numerous prior works. Regarding the scaling issue with $\sqrt{\theta}$, we assumed a normalized model to avoid such ambiguities, ensuring that our results remain valid and interpretable. We will add a note in the manuscript to clarify this assumption.
>
> ## **Relationship with Transitional Regimes**
>
> As noted earlier, **we validated our theoretical predictions for $d=5{,}000$ and observed consistency with empirical results**. While these findings suggest robustness in transitional regimes, determining the precise applicability of our analysis to smaller $d$ remains an open question and an intriguing direction for future work.
>
> ## **Datasets with Complex Spectrums**
>
> The spiked covariance model is a standard framework in theoretical analyses of unsupervised learning discussed in RELATED WORK (High-dimensional asymptotics from replica method ), and denoising autoencoders [2], autoencoders [3].
> **Prior studies have demonstrated Gaussian universality, showing that Gaussian models can effectively explain the behavior of complex real-world datasets, as discussed in Line 197-201 and Line 494-498.**
> It has also been reported that Gaussian Universality exists in autoencoders where the VAE has been made deterministic [3].
> However, these previous studies also used datasets such as MNIST, FashionMNIST, CIFAR10, and ImageNet, leaving it uncertain whether Gaussian Universality holds in cases with excessively complex spectrums.
>
> ## **Clarification on Equation (9)**
>
> To address your question about Equation (9), we have revised the manuscript to explicitly state that the expectation is taken over the data distribution. We hope this clarification resolves any ambiguity regarding the notation.
>
> By addressing your concerns about model simplicity, transitional regimes, and dataset complexity, we believe we have strengthened the paper and clarified its contributions.  We kindly request that you reconsider the scores for Soundness and Contribution in light of our revisions.

---

> > ### Comment · Reviewer_1vWD · 2024-11-24
> >
> > Thank you for these comments.
> > The identifiability of a statistical model is central when you try to estimate the parameters (see any basic statistics course). Without identifiability, this task is ill posed. Although it appears in the related works, I would have prefered to see at least a comment on this model.
> > In any case, I feel this work is interesting but still at the beginning of its potential impact with many other aspects to analyse. I keep my score unchanged.

---

> > > ### Author Response · Authors · 2024-11-28
> > > **Response**
> > >
> > > Thank you for your response and valuable feedback.
> > > We appreciate the opportunity to clarify and address the points you raised. Below, we respond to your concerns in detail and outline the steps we have taken to improve our submission based on your comments.
> > >
> > > ## **Regarding Identifiability and Ill-Posedness**
> > >
> > > We understand that you are raising concerns about the *identifiability* of the statistical model in our problem setting.
> > > However, we would greatly appreciate it if you could elaborate on the specific reasons why you believe our problem setting might be ill-posed.
> > >
> > > For instance, while it is true that in cases where $k^{\ast}<k$, multiple candidate models of Linear VAEs can reproduce the statistical properties of the data, would any of these models still result in successful training?
> > > This suggests that the lack of strict *identifiability* may not hinder the overall utility of the setting.
> > > If there is a specific scenario or example where this ambiguity leads to a meaningful failure, we would be eager to analyze it and incorporate the findings into the Camera-Ready version.
> > >
> > > Furthermore, **we are unclear about your reference to dividing by $\sqrt{\theta}$.** If you could provide more details or clarify this point, we would be happy to include a corresponding explanation or additional analysis in the revised version.
> > >
> > > Finally, in the specific setting where $k=k^{\ast}=1$, *identifiability issues* might not arise. We believe that the results we demonstrated—such as the inevitable posterior collapse and the double descent phenomenon in VAEs—are meaningful contributions that remain valid regardless of identifiability concerns in higher dimensions.
> > >
> > > We appreciate that you acknowledged our response to your initial comment on the intermediate regime and that it has adequately addressed your concern. **Building upon this, we have further extended one of our claims to the $k>1$ case, as noted in Lines 456–460 of the revised manuscript.** This addition represents our efforts to strengthen the paper based on your valuable feedback, and we hope this extension provides further clarity and rigor to our contributions.
> > >
> > > In light of the clarifications above, the additional analyses we conducted, and the revisions we have made to the manuscript, we kindly request that you reconsider your score. We sincerely hope that these efforts address your concerns.

---

> ### Author Response · Authors · 2024-12-03
> **Thanks**
>
> Dear Reviewer 1vWD,
>
> We greatly appreciate the time and effort you have dedicated to reviewing our work. As the deadline approaches with only one day remaining, we sincerely request your feedback on our rebuttal. Please inform us if any aspect of our explanation remains unclear.
>
> We would greatly appreciate your confirmation on whether your concerns have been adequately addressed. If the issues are resolved, we would appreciate your reevaluating this study. We will respond promptly before the discussion deadline if further clarification is required.
>
> Best,
>
> The authors

---

### Official Review · Reviewer_Xf1j · 2024-11-02

**Soundness:** 3
**Presentation:** 3
**Contribution:** 3
**Rating:** 6
**Confidence:** 4

**Summary:**

This paper aims to analyze the solution learnt by a $\beta$-VAE w.r.t. (1) the parameter $\beta$, and (2) the training dataset size. The authors work in the high-dimensional asymptotic setting, and aim to characterize certain phenomena about the quality of the learnt solution by the VAE.

To do this, the authors theoretically analyze a linear VAE model (Eq (5)), in the high-dimensional asymptotic regime where $n,d \rightarrow \infty$ and $\frac{n}{d} = \alpha$ (sample complexity) stays finite, using the replica method as a heuristic to get around intractable calculations. This is presented in Section 5.

The asymptotic formulae are empirically verified in Section 6.1 and 6.2, on a synthetic data model of the spiked covariance matrix (Eq (4)). This is then used to draw interesting observations about the learning process and the quality of the learnt VAE solution. Figure 2 in particular shows many of the findings.

The authors then empirically show some of the findings (from the linear VAE setting) hold true for non-linear VAEs also, trained on real-world datasets like MNIST and FashionMNIST. This is presented in Section 6.5.

**Strengths:**

**Technical strengths**:
- The paper sharply characterizes high-dimensional asymptotics for learning the linear VAE (Eq (5)) under the spiked covariance model (Eq (4)) with the regularized $\beta$-VAE objective (Eq (6)).
- This is used to show interesting observations about the VAE learning process in Section 6.1 and 6.2. In particular, (1) Figure 2 shows a double-descent phenomenon w.r.to the sample complexity $\alpha$, with the reconstruction error (Eq (9)) peaking at $\alpha = 1$, and (2) Figure 2 also shows a long plateau in the reconstruction error for large values of $\beta$. This is backed by Claim 6.1 (in the large $\alpha$ limit) and lines 428-430 provide concrete guidance to practitioners about the risks of a large $\beta$ when training.
- Section 6.5 (and Figure 5) shows this on real-world datasets MNIST and FashionMNIST also, where the insight can be used to practically choose the "optimal" value of $\beta$ approximately equal to the noise ratio $\hat{\eta}$, which can be estimated using the training dataset.

**Presentation strengths**:
- The paper is largely well-written and easy to follow. The authors include relevant explanations in most places. For example, the choice of the spiked covariance model as the synthetic data generating process was backed by evidence in Figure 1 of MNIST following something similar.

**Weaknesses:**

**Technical Weaknesses**:
- The main weakness is the fact that the theoretical results are not exact, since they have been developed using the replica method, which is a heuristic to get around intractable calculations.
- The authors work in the simple setting of $k = k^\star = 1$. If I understand correctly, this means the true latent space is $1$-dimensional. It would have been nice to see the synthetic experiments with $k^\star$ varying, say in $[1, 2, 4]$. In particular, what would the trend of $\varepsilon_g$ w.r.to $k^\star$ look like?
- Some of the claims can be better substantiated. For example, in the context of Figure 2, it would have been nice to see a plot of $\varepsilon_g$ w.r.to $\alpha$ for the optimal $\beta$ choice. Is that perhaps monotonically decreasing? (This is similar to the double-descent observations in literature, where using the optimal parameter leads to a monotonically decreasing curve instead of double-descent).

**Minor notes on the typos I found**
- Line 146, $H$ is probably the "negative log-likelihood" instead of just "likelihood". I stress this because it is important whether we want to minimize or maximize $H$.
- Line 189, "Spectrum" of the covariance matrix, instead of "Spectral". This typo is present in many places throughout the paper (for eg, Fig 1(b)), would appreciate if it can be cleaned up.
- Line 214, "Note that" instead of "Noted that".
- Line 224, $\lambda \in \mathbb{R}_{+}$ maybe instead of  $\lambda \in \mathbb{R}$? I *assume* practitioners use a non-negative regularization parameter.

**Questions:**

- What is the main challenge that the replica method allows you to get around? Would be nice to provide some insight into this. Or perhaps a toy example of the usage of replica method demonstrating what are its benefits and why is it used in this particular context.
- What is the main reason to introduce the metric $\varepsilon_g$ in Eq (9)? How is it different than the distortion $D$?
- In Figure 3, what does "overlearning" mean? From the description in section 6.2, it seems it is the same as overfitting? If so, would be good to name it that way instead of introducing a new term.

---

> ### Author Response · Authors · 2024-11-21
> **Response (1)**
>
> We sincerely thank you for your thoughtful and detailed review.
> We are especially grateful for your recognition of our contributions, including identifying the double-descent phenomenon, the inevitable posterior collapse, and their practical implications.
> In the following, we address your comments and questions in detail.
>
> ## **Replica Method**
>
> As noted in the RELATED WORK section, the replica method is a well-established tool for theoretical analyses in high-dimensional statistics and has been successfully used to explain various phenomena in machine learning [1–3], including denoising autoencoders [4] and autoencoders [5].
> Recent studies have increasingly demonstrated its mathematical rigor, as discussed in the revised Related Work section (High-Dimensional Asymptotics from the Replica Method, Line 113-117).
>
> In our study, the replica method was essential for deriving sharp predictions regarding the dataset-size dependence of the signal recovery error and RD curve behavior. **These theoretical predictions were validated through numerical experiments with $d=5{,}000$**. As shown in Figure 2, the strong agreement between the numerical results for finite $d=5{,}000$ and the predictions from the replica method provides strong evidence of its validity in this context. This alignment also highlights the robustness of the analysis in the high-dimensional limit where $d \to +\infty$ and $n \to +\infty$ with a fixed ratio $\alpha = d/n$ for the regime where $d$ and $n$ are finite.
>
> - [1] Lenka Zdeborova, Insights from exactly solvable high-dimensional models, ICLR2023
> - [2] Cory Stephenson et al., On the geometry of generalization and memorization in deep neural networks, ICLR2021
> - [3] Federica Gerace et al., Generalisation error in learning with random features and the hidden manifold model, ICML 2020
> - [4] Hugo Cui and Lenka Zdeborova, High-dimensional Asymptotics of Denoising Autoencoders, NeurIPS2023
> - [5] Maria Refinetti and Sebastian Goldt, The dynamics of representation learning in shallow, non-linear autoencoders, ICML2022
>
> ## **On the Simple Setting**
>
> We appreciate your observation regarding the simplicity of our setting ($k=1$).
> **Our primary contribution is the derivation of a general formula (Claim 4.2), enabling a sharp analysis of how various metrics in linear VAEs- such as rate $R$, distortion $D$, signal recovery error $\varepsilon_{g}$, $D_{\mathrm{KL}}[p(x) \| p_{\mathrm{data}}(x)]$**.
> While we focus on the simplest case with $k=k^{\ast}=1$, this minimal case still captures critical phenomena such as **nevitable posterior collapse** and the **double-descent behavior**.
> Similar minimal settings with $k=1$, have been the focus of analyses on denoising autoencoders [4].
> In situations where $k^{\ast} > 1$, as you pointed out, a phenomenon similar to the *progressive learning* of strong spikes, observed in the analysis of autoencoders [5], may emerge. A detailed analysis of this scenario is left as part of our future work.
>
> Additionally, if you could share specific references for [1], [2], and [4], it would significantly enhance our understanding.
>
> ## **Optimal $\beta$**
>
> As mentioned in Line 376-405 of the manuscript, the optimal $\beta_{\mathrm{VAE}}$ depends on the sample complexity $\alpha$, which behaves quantitatively differently from the ridge regularization strength $\lambda \in \mathbb{R}_{+}$.
>
> Figure 2 (Left) demonstrates the effect of varying the regularization strength $\lambda$, while Figure 2 (Middle) shows the effect of varying $\beta_{\mathrm{VAE}}$.
> **This $\alpha$-dependence of optimal $\beta_{\mathrm{VAE}}$ is a key insight for practitioners**.
> When $\alpha \to \infty$, setting the optimal $\beta_{\mathrm{VAE}}=\eta$ still results in the double-descent phenomenon.
>
> ## **Minor notes**
>
> We sincerely appreciate your attention to detail and have made the necessary corrections in the revised manuscript.

---

> > ### Comment · Reviewer_Xf1j · 2024-11-23
> > **Thank you for the response**
> >
> > Thank you for the detailed clarifications you provided.
> >
> > Regarding my comment on the $k = k^\star = 1$ (simple setting), I think there is a small misunderstanding. I did not mean [1,2,4] as references to other papers. What I meant was you should also consider providing experimental results for $k = k^\star = 2,4$ also instead of just fixing latent dimension to $1$.
> >
> > Overall, I will retain my score.

---

> > > ### Author Response · Authors · 2024-11-27
> > > **Response**
> > >
> > > We sincerely appreciate your valuable feedback and the opportunity to address your concerns.
> > >
> > > One of the central claims of our study is the existence of "posterior collapse" that cannot be avoided by increasing the dataset size.
> > > In response to your comment, we have verified whether this phenomenon holds for any given $k = k^{\ast}$.
> > > Revised manuscript shows that the "inevitable posterior collapse" occurs for arbitrary $k = k^{\ast}$. Furthermore, the threshold condition is consistent with the case of $k = k^{\ast} = 1$, where $\beta_{\mathrm{VAE}} = \rho + \eta$.
> > > **This result has been incorporated into the revised version of our manuscript (Lines 457--460), with detailed proof provided in the Appendix (Lines 1134--1167).**
> > >
> > > Additionally, we would like to clarify the experimental settings for MNIST, FashionMNIST, and CIFAR10.
> > > As detailed in Appendix E: EXPERIMENT DETAILS, we did not use models with a one-dimensional latent variable. Specifically, for MNIST and FashionMNIST, we utilized models with two-dimensional latent variables, while for CIFAR10, models with 128-dimensional latent variables were employed.
> > > **Even with these configurations, our experiments consistently revealed regions where the FID score does not improve with increasing dataset size, aligning well with the predictions of our theoretical analysis.**
> > >
> > > We hope these clarifications address your concerns effectively. We believe this strengthens the connection between our theoretical insights and practical applications of VAEs in real-world engineering contexts.
> > >
> > > Thank you again for your constructive feedback and support.

---

> ### Author Response · Authors · 2024-11-21
> **Response (2)**
>
> ## **Usage of Replica Method**
>
> The replica method provides significant advantages for analyzing high-dimensional learning models.
> Traditional PAC-bound analyses of generalization error often assume that the data are iid and focus on worst-case scenarios.
> These approaches fail to explain modern phenomena, such as the double-descent effect, in which increasing model capacity improves generalization after the interpolation peak.
> Furthermore, replica analyses enable sharp evaluations of the dataset-size dependence of generalization error by incorporating data structures and architectural features into the analysis [1,2,3,4,5].
> Indeed, as in other replica analyses of unsupervised learning discussed in RELATED WORK (High-dimensional asymptotics from the replica method), we assume a spiked data structure, which enables us to characterize phenomena such as posterior collapse and double descent.
> This flexibility and precision are key strengths of the replica method.
>
> ## **Clarifications on Metrics (Eq. 9)**
>
> We appreciate your question about the distinction between signal recovery error and the distortion metric. These metrics serve different purposes, as clarified in the revised manuscript (Lines 222–224):
>
> - **Distortion (Reconstruction Error)**: : Measures how well the data can be reconstructed after compression into and decoding from the latent space, reflecting the fidelity of the VAE’s encoder-decoder process.
> - **Signal Recovery Error**: Focuses on the decoder alone and evaluates how well the latent variables $c$ are decoded into the data space. It provides a measure of how closely the data generated by the VAE matches the true data distribution, independent of the encoder.
>
> ## **overlearning**
>
> To avoid confusion, we have replaced *overlearning* with *overfitting* in the revised manuscript, aligning with standard terminology.
>
> By incorporating your feedback on the replica method, simple settings, optimal $\beta_{\mathrm{VAE}}$, and distinctions between metrics, we believe the revised manuscript more effectively conveys our contributions and their broader implications for the machine learning community. We respectfully request that you reconsider the scores for Soundness and Contribution based on these revisions.

---

> ### Author Response · Authors · 2024-12-03
> **Thanks**
>
> Dear Reviewer Xf1j
>
> We greatly appreciate the time and effort you have dedicated to reviewing our work. As the deadline approaches with only one day remaining, we sincerely request your feedback on our rebuttal. Please inform us if any aspect of our explanation remains unclear.
>
> We would greatly appreciate your confirmation on whether your concerns have been adequately addressed. If the issues are resolved, we would appreciate your reevaluating this study. We will respond promptly before the discussion deadline if further clarification is required.
>
> Best,
>
> The authors

---

### Official Review · Reviewer_LZGQ · 2024-11-04

**Soundness:** 3
**Presentation:** 2
**Contribution:** 3
**Rating:** 8
**Confidence:** 3

**Summary:**

The paper investigates several aspects of beta-linear-VAEs, including posterior collapse, the effect of different values for the beta parameters, the effect of training dataset size.
It also introduced two summary statistics, and using these statistics, the authors derived a phase diagram for VAE learning in terms of the beta values and the relative scale of the training dataset size.

**Strengths:**

- The paper studied an important aspect of VAEs and how the different parameters and choices can affect the performance.
- The empirical findings of the relation between generalisation error and the sample complexity as well as the beta parameter is interesting.

**Weaknesses:**

The paper discussed a list of different behaviours of VAEs, but it feels like they are rather loosely connected findings (i.e., the subsections in Section 6).

The findings themselves are interesting, but it is not surprising that changing one variable, such as beta or the number of training data, will lead to various changes in aspects like RD curves, posterior collapse.

Therefore, I believe a more coherent story is important to connect the dots and make these findings more insightful.

**Questions:**

1. The signal recovery error feels like a definition of reconstruction error, and a distortion metric. What’s the difference between the signal recovery error and the distortion (D) of the RD curve in the paper?
2. Typo: Page 8 line 397: “summary statics” -> “summary statistics”

---

> ### Author Response · Authors · 2024-11-21
> **Response to Weaknesses**
>
> We sincerely appreciate your thoughtful review and constructive suggestions.
> Your comments have been instrumental in helping us identify opportunities to improve the clarity, coherence, and practical relevance of our manuscript. Below, we provide detailed responses to each of your points.
>
> ## **Weakness: Coherence of Findings**
>
> If we understood your concern correctly, it relates to the possibility that the results in Section 6.5, where we compare theoretical predictions with experiments on real-world data and nonlinear VAEs, might appear loosely connected.
> You are correct that our theoretical results do not guarantee exact alignment with real-world data or more complex models. **However, our main contribution of this paper is the derivation of a general formula, presented in Claim 4.2, for analyzing the dependence of sample complexity $\alpha$ on VAEs**.
>
> The previous studies in the high-dimensional limit where $d \to +\infty$ and $n \to +\infty$ with a fixed ratio $\alpha = d/n$, cited in the main text (Lines 499–501 in Section 6.5, Line 196-201 in Section 4) and [1], that examines the dependence of sample complexity $\alpha$ typically focus only on MNIST and CIFAR10 datasets. These studies support the Gaussian universality [1], which suggests that the data generation process of a Gaussian model can explain real-data phenomena.
> Accordingly, we employed this setting and considerations for our numerical experiments on MNSIT, FashionMNIST, and CIFAR10.
> While the studies in the high-dimensional limit where $d \to +\infty$ and $n \to +\infty$ with a fixed ratio $\alpha = d/n$ needs to verify theoretical consistency across more comprehensive datasets, this lies beyond the scope of the current work. Our main contribution is theoretical analysis, and we consider more exhaustive empirical validation a significant step in future work.
>
> - [1] Maria Refinetti and Sebastian Goldt, The dynamics of representation learning in shallow, non-linear autoencoders, ICML2022
>
> ## **Connecting the Theoretical Findings**
>
> We agree that posterior collapse for high $\beta_{\mathrm{VAE}}$ might not be surprising.
> **However, as Reviewer Xf1j noted, "Figure 2 also shows a long plateau in the reconstruction error for large values of $\beta$ . This is backed by Claim 6.1 (in the large $\beta$ limit) and lines 428-430 provide concrete guidance to practitioners about the risks of a large $\beta$ when training."**
> In response to your suggestion, **we revised CONCLUSION section (Line 516-529) to explicitly summarize the practical implications of our findings**, emphasizing the tuning of $\beta_{\mathrm{VAE}}$ for engineering applications of VAEs. This revision links our theoretical insights to real-world applications, bridging the gap between abstract findings and practical implementation.
>
> ## **Signal Recovery Error vs. Reconstruction Error**
>
> We have clarified this in the revised manuscript (Lines 244–250).
> To summarize:
>
> - **Distortion (Reconstruction Error)**: Measures how well the data can be reconstructed after compression into and decoding from the latent space, reflecting the fidelity of the VAE’s encoder-decoder process.
> - **Signal Recovery Error**: Focuses on the decoder alone and evaluates how well the latent variables $c$ are decoded into the data space. It provides a measure of how closely the data generated by the VAE matches the true data distribution, independent of the encoder.
>
> We believe these revisions address your concerns and significantly improve the coherence, clarity, and practical impact of our paper. We kindly ask you to reconsider your evaluation of our work in light of these improvements.

---

> ### Comment · Reviewer_LZGQ · 2024-11-24
>
> I would like to thank the authors for their detailed rebuttal and clarifications. My questions are resolved. I will keep my score.
>
> I believe to take this work further with current contributions (which are valuable) might require a bit more rewriting at the higher level. This will help to make the `core message` clearer, as also mentioned by Reviewer NKDk.

---

> > ### Author Response · Authors · 2024-11-28
> > **Response**
> >
> > Thank you for your valuable feedback and for taking the time to review our rebuttal. We are pleased to hear that the concerns raised have been resolved, and we appreciate your constructive suggestions.
> >
> > **In response to your insightful comments, we have extended one of our key claims to the case of $k > 1$, as detailed in Lines 456–460 of the revised manuscript. This extension enhances the practical implications of our findings and better connects our theoretical contributions to real-world engineering applications.**
> > Additionally, we recognize the importance of clearly articulating the relevance of our results for engineering applications. As a step in this direction, we have added concise interpretations in Section 6 for each result, and we plan to further elaborate on these implications in the camera-ready version.
> >
> > We believe these improvements significantly strengthen the clarity and impact of our work. Considering these updates, we kindly ask you to reconsider your score, as we hope the revised version aligns more closely with your expectations for both theoretical contributions and their applicability.

---

> > > ### Comment · Reviewer_LZGQ · 2024-11-28
> > >
> > > Thank you for the updates. Even though I still believe the current manuscript can be written better and clearer, the contributions are indeed valuable. I will change my score from 6 -> 8.
> > >
> > >
> > > But I do understand the concerns from other reviewers, and I hope they can be addressed well.

---

### Official Review · Reviewer_NKDk · 2024-11-04

**Soundness:** 2
**Presentation:** 3
**Contribution:** 1
**Rating:** 5
**Confidence:** 4

**Summary:**

This paper studies the RD curves in VAEs from a function of dataset size and dimensionality. The authors suggest that the RD curves as a function of data complexity $\alpha$ (# data points / dim of data) and $$\beta, can be divided to three categories of overfitting, learning, and underfitting; In high $\alpha$ regime, smaller $\beta$ is needed in order to avoid over-regularizing the model.

**Strengths:**

- To the best of my knowledge, this is the first paper that studied RD curves in VAEs as a function of dataset size and data dimensions. This topic I think is a valuable topic of study and will indeed be of interest to the ICLR community.

- The theory in the paper, to the best of my understanding, is sound.

- The paper for the most reads well.

**Weaknesses:**

- There is no study of the network capacity in this work. While I understand that this is theoretical work, the authors do make a claim that the same results hold for more complex networks. However, there are prior works that suggest that RD curves for different network capacities behave differently [1,2]. Could the authors comment on this?

- It is also not clear to me what is the message of the paper. It ofcourse makes sense that when you don't have a lotta data in high dimensions, you want to incorporate prior knowledge (such as regularization). Similarly,  when you have a lotta data, you don't need a lotta regularization as it is evident from all the recent DGMs. Furthermore, the $\alpha < 1$ is hardly interesting as it is almost never the case. So practically, what does this mean for people employing VAEs? What is the core message here?

- I do not think the experiments are strong enough to back the claims made in this paper. First, $\alpha$ should have been studied as a function of both $n$ and $d$ separately. Here, $d$ was kept fixed. Furthermore, the data choice here is extremely specific. I understand the design choice, but some controlled experiments on real-world datasets are also necessary before showing Figure 3 left with those specific values.


[1] Bozkurt, Alican, et al. "Rate-regularization and generalization in VAEs." arXiv preprint arXiv:1911.04594 (2019).

[2] Chérief-Abdellatif, Badr-Eddine, et al. "On PAC-Bayesian reconstruction guarantees for VAEs." International conference on artificial intelligence and statistics. PMLR, 2022.





**Minor comments**

- "Notations " should not be place in Related work I would say
- I would strongly advise to avoid using $D$ for the variances in $q$ and use $\sigma^2$ instead as it is the most common symbol in the literature  for this.

**Questions:**

- Can you comment on how much the analysis is effected by the fact that $D$ is fixed?
- Are the RD curves computed for the training set or test set? These two curves can be widely different.
- Do the authors mean overfitting by "overlearning"? If yes, I would say replace it with overfitting to avoid confusion :)
- this is not clear to me but how did the authors at the values for Figure 3 Left? This is not seem to match other figures.

---

> ### Author Response · Authors · 2024-11-21
> **Response (1)**
>
> We sincerely appreciate your detailed and constructive feedback. We are encouraged by your acknowledgment of our work's relevance to the ICLR community, especially in contributing to the theoretical understanding of the dataset-size dependence of RD curves. Below, we address your comments in detail.
>
> ## **Relation to [1, 2]**
>
> If we are not mistaken, there appears to be a misunderstanding regarding the results presented in Section 6.5.
> Section 6.5 of our manuscript investigates how the signal recovery error depends on $\beta_{\mathrm{VAE}}$, showing that the error demonstrates qualitatively consistent behavior across varying network capacities. This consistency confirms the inevitable posterior collapse and highlights consistent trends in optimal $\beta_{\mathrm{VAE}}$ corrections for finite $\alpha$.
> Note that the RD curve analysis in Section 6.4 is purely theoretical and independent of network capacity considerations.
>
> In the following, we explore the relationship between references [1] and [2].
> For reference [1], although variations in network capacity influence RD curve behavior, the qualitative trends we predict remain consistent.
> For example, Figure 4 in [1] illustrates RD curves moving closer to the rate and distortion axes as the dataset size increases, which aligns with our theoretical prediction.
> Similarly, the gradient of $-1$ observed around $\beta_{\mathrm{VAE}}=1$ supports this consistency. Additionally, Figure 4 in [1] provides numerical evidence that larger datasets are necessary in high-rate regions, aligning with our theoretical predictions.
> We acknowledge that further analysis of the relationship between network capacity and RD behavior is significant for future research.
> In reference [2], the PAC-bound analysis offers a worst-case scenario, and the correspondence between their results and ours is not immediately evident.
>
> - [1] Bozkurt Alican et al., Rate-regularization and generalization in VAEs, arXiv preprint arXiv:1911.04594 (2019).
> - [2] Cherief-Abdellatif Badr-Eddine et al., On PAC-Bayesian reconstruction guarantees for VAEs, AISTATS2022.
>
> ## **Core Message and Practical Implications**
>
> We appreciate the feedback regarding the clarification of our core message.
> **Our primary contribution lies in the theoretical characterization of VAEs under high-dimensional asymptotics, where $d \to +\infty$ and $n \to +\infty$ with a fixed ratio $\alpha = d/n$**.
> This approach has gained attention in learning theory because this regime captures intriguing phenomena such as double descent and the advantages of low-dimensional manifold structures [3, 4, 5], which cannot be explained by traditional PAC-bound methods.
> Recent studies have applied this framework to denoising autoencoders [6] and standard autoencoders [7]. In this work, we extend the framework to VAEs and derive a general formula (Claim 5.2) to analyze the dataset-size dependence of generalization performance and posterior collapse.
>
> A key practical insight from our analysis is the importance of understanding and tuning
> $\beta_{\mathrm{VAE}}$.
> **As noted by Reviewer Xf1j, "Figure 2 also shows a long plateau in the reconstruction error for large values of $\beta$. This is backed by Claim 6.1 (in the large $\beta$ limit) and lines 428-430 provide concrete guidance to practitioners about the risks of a large $\beta$ when training."**
> We will revise the manuscript to highlight these insights and their relevance to real-world applications, summarizing **the key engineering takeaways in the updated CONCLUSION section (Line 516-529).**
>
>
> - [3] Lenka Zdeborova, Insights from exactly solvable high-dimensional models, ICLR2023
> - [4] Cory Stephenson et al., On the geometry of generalization and memorization in deep neural networks, ICLR2021
> - [5] Federica Gerace et al., Generalisation error in learning with random features and the hidden manifold model, ICML 2020
> - [6] Hugo Cui and Lenka Zdeborova, High-dimensional Asymptotics of Denoising Autoencoders, NeurIPS2023
> - [7] Maria Refinetti and Sebastian Goldt, The dynamics of representation learning in shallow, non-linear autoencoders, ICML2022
>
> ## **The regime $\alpha < 1$**
>
> The regime $\alpha < 1$, where the dimensionality of the data exceeds the sample size, is uncommon in standard benchmark datasets but plays a significant role in high-resolution or low-sample-size scenarios.
> This regime is essential for understanding overparameterization and its relationship to the phenomenon of double descent.
> **Although our analysis encompasses the regime $\alpha < 1$, it is not restricted to this range; Claim 4.2 allows for a comprehensive characterization of signal recovery error and RD behavior over any $\alpha$, ensuring its broad applicability.**

---

> ### Author Response · Authors · 2024-11-21
> **Response (2)**
>
> ## **Experiment**
>
> **The primary contribution of this paper is the derivation of a general formula, presented in Claim 4.2, for analyzing the dependence of sample complexity on VAEs**.
> To validate our theoretical predictions, we conducted numerical experiments using the widely studied MNIST and CIFAR10 datasets. These datasets are commonly used in previous studies, cited in the main text (Lines 499–501 in Section 6.5, Line 196-201 in Section 4) and [7], which support the Gaussian universality hypothesis [7]. **This hypothesis suggests that the data generation process of a Gaussian model can explain fundamental phenomena observed in real-world data.**
> While we recognize the significance of verifying theoretical consistency across more comprehensive datasets, this lies beyond the scope of the current work. Our focus is on establishing a foundational theoretical framework, which we believe is essential for guiding future empirical validations.
>
> ## **Minor Comments**
>
> We have moved the ``Notations'' section to precede Related Work for improved clarity. Using $D$ for encoder variance follows conventions in prior studies summarized in Linear VAEs in RELATED WORK and is standard in theoretical analyses of linear VAEs.
>
> ## **Fixed $D$**
>
> While $D$ was fixed to isolate the effects of $\beta_{\mathrm{VAE}}$, it can also be treated as an optimizable parameter, as demonstrated in [8].
>
> - [8] N. Barkai and H. Sompolinsky, Statistical Mechanics of the maximum-likelihood density estimation, Physical Review E 50.3 (1994): 1766.
>
> ## **RD Curve Evaluation**
>
> The RD curves shown in Figure 4 represent theoretical results based on Claim 6.2 rather than being derived from empirical test data.
>
> ## **Overlearning**
>
> To avoid confusion, we have replaced *overlearning* with *overfitting* throughout the manuscript, as suggested.
>
> ## **Figure 3 Explanation**
>
> Figure 3 presents a phase diagram demonstrating the detailed state of linear VAEs based on the parameters $Q$ and $m$, providing a comprehensive perspective on the generalization error trends shown in Figure 2.
> For instance, in Figure 2 (middle), at $\beta_{\mathrm{VAE}} = 1.0$ with $\alpha = 2$, the signal recovery error begins to decrease, corresponding to the *Learning Phase* in the phase diagram.
> In contrast, at $\beta_{\mathrm{VAE}} = 1.8$ and $\alpha = 1.5$, the signal recovery error does not improve, aligning with the *Regularized Phase* in the phase diagram. Detailed quantitative distinctions between these phases are discussed in Section 6.2.
>
>
> We hope our responses have sufficiently addressed your questions and clarified the contributions of our work. We kindly request that you reconsider your evaluation of our submission.

---

> > ### Comment · Reviewer_NKDk · 2024-11-25
> > **Respond to the Rebuttal**
> >
> > Thank you for your response. I've read the rebuttal as well as the other reviews. The consensus does seem to be that there is a gap between some of the claims and the experiments, as well as the significance of the contributions.
> >
> > My questions regarding the RD curves and fixed D has been addressed. However, the core issue remains the same. The contributions still seem misguided to me. As I mentioned, figure 3 (left) I believe does not add much to our understanding of VAEs as it matches the general ML intuition.  Figure 3 (right) does make sense but this is also not surprising that for easier problems, you get better rate and distortions. Furthermore, the study is undermined by the practical verified difference between training and test RD values, as well as the different shapes of RD curves for different network capabilities. Sorry in advance if I'm misunderstanding but  I do not follow the argument of "long plateau in the reconstruction" for high $\beta$ values. Higher $\beta$ leads to worse reconstructions, which is what Figure (2) middle showing. What am I missing here?
> >
> > Overall, while I think there is value to the theoretical findings of the paper, I keep my score.

---

> > > ### Author Response · Authors · 2024-11-28
> > > **Response**
> > >
> > > Thank you for your thoughtful and detailed feedback.
> > > We appreciate the opportunity to clarify the points you raised and address your concerns regarding the theoretical and experimental contributions of our work.
> > >
> > > ## **On the Gap Between Theoretical Analysis and Numerical Experiments**
> > >
> > > We are somewhat uncertain about your comment and would greatly appreciate it if you could clarify which specific aspects of the gap between the numerical experiments and theoretical analysis you are referring to.
> > > We would like to ensure that there is no gap regarding the alignment between our theoretical analysis and numerical experiments.
> > > As demonstrated in Section 6.5, the signal recovery error behavior derived theoretically matches the qualitative trends observed in our numerical experiments.
> > > Furthermore, as you referenced [1], we note that the qualitative behavior of RD curves in [1] aligns with our theoretical results.
> > >
> > > If your concern lies in the lack of empirical verification of RD curves in real-world scenarios similar to [1], we are open to including additional numerical validation in the Camera Ready version.
> > > While we acknowledge the importance of extending our analysis to more complex deep learning models, such a task remains inherently challenging in current learning theory, even for supervised learning.
> > > **As a first step, our focus on the data-dependence of RD curves and signal recovery error using a minimal model (Linear VAE) provides a foundational contribution that we believe is crucial for advancing this line of research.** We hope you will consider the value of this incremental yet significant step toward understanding the principles governing VAEs.
> > >
> > > - [1] Bozkurt Alican et al., Rate-regularization and generalization in VAEs, arXiv preprint arXiv:1911.04594 (2019).
> > >
> > > ## **On the Importance of Figure 3**
> > >
> > > We strongly believe that Figure 3 plays a central role in elucidating the behavior of Linear VAEs and represents the core contribution of our paper.
> > > This phase diagram offers a complete description of Linear VAE behavior in the sample complexity $\alpha$ and $\beta_{\mathrm{VAE}}$ space. Importantly, it goes beyond prior numerical studies, which largely emphasize posterior collapse is related to $R \approx 0$, by revealing that regions inducing posterior collapse correspond to distinct phases— the *Overfitting Phase* and the *Regularized Phase*, as explained in Section 6.2.
> > >
> > > Moreover, Figure 3 identifies the sharp boundaries between these phases in the $\beta_{\mathrm{VAE}}$–$\alpha$ space, providing insights that we believe are non-trivial. For instance, the origins of posterior collapse differ fundamentally between these two phases, which offers a deeper understanding of the phenomenon. This structured characterization advances the discussion around VAEs beyond empirical observations, contributing new insights that we hope the community will find valuable.
> > >
> > > ## **On the Long Plateau Phenomenon**
> > >
> > > We apologize for any confusion regarding the "long plateau" discussion. **The term "long plateau" is specifically used for the signal recovery error $\varepsilon_{g}$, not the reconstruction error (distortion)**. The signal recovery error, as explained in Section 6.2, measures the discrepancies between the decoder's learned and true data distributions, thus distinguishing it from distortion.
> > >
> > > As shown in Figure 2 (Middle), for large $\beta_{\mathrm{VAE}}$ values (e.g., $\beta_{\mathrm{VAE}} = 1.8, 2.0$), $\varepsilon_{g}$ exhibits a long plateau at $\varepsilon_{g} \approx 1$.
> > > Furthermore, in Figure 3, we identify that as $\alpha \to \infty$, the boundary between the *Regularized Phase* and the *Learning Phase* asymptotically approaches $\beta_{\mathrm{VAE}} = \rho + \eta$.
> > > This asymptotic behavior explains the long plateau phenomenon: as $\beta_{\mathrm{VAE}}$ approaches this boundary from below, the plateau length increases indefinitely.
> > > **This finding provides not only a theoretical understanding of posterior collapse for large $\beta_{\mathrm{VAE}}$, but also engineering insights for designing VAEs, as highlighted in Section 7 (Lines 518–532).**
> > >
> > >
> > > To better address the relevance of our findings to engineering applications, we have generalized one of our claims in the revised version (Lines 456–460).
> > > While we understand your emphasis on practical applications, we ask you to consider our contributions in the context of the ICLR community’s growing interest in theoretical perspectives on representation learning. Our work aims to provide a long-term theoretical foundation for such applications, and we hope this alignment with the community's objectives will lead you to reevaluate your score.
> > >
> > > Your constructive comments have been invaluable in refining our paper, and we hope this response provides clarity and adequately addresses your concerns. Thank you for your thoughtful feedback.

---

> ### Author Response · Authors · 2024-12-03
> **Thanks**
>
> Dear Reviewer NKDk
>
> We greatly appreciate the time and effort you have dedicated to reviewing our work. As the deadline approaches with only one day remaining, we sincerely request your feedback on our rebuttal. Please inform us if any aspect of our explanation remains unclear.
>
> We would greatly appreciate your confirmation on whether your concerns have been adequately addressed. If the issues are resolved, we would appreciate your reevaluating this study. We will respond promptly before the discussion deadline if further clarification is required.
>
> Best,
>
> The authors

---

### Author Response · Authors · 2024-12-02
**Thanks**

Dear Reviewer NKDk,  Xf1j, 1vWD,

We greatly appreciate the time and effort you have dedicated to reviewing our work.
As the deadline approaches with only few hours remaining, we sincerely request your feedback on our rebuttal.
Please inform us if any aspect of our explanation remains unclear.

We would greatly appreciate your confirmation on whether your concerns have been adequately addressed.
If the issues are resolved, we would appreciate your reevaluating this study.
We will respond promptly before the discussion deadline if further clarification is required.

Best,

The authors

---

> ### Author Response · Authors · 2024-12-03
> **Response**
>
> Dear Reviewer NKDk, Xf1j, 1vWD,
>
> Thank you for the time and effort you have dedicated to reviewing our work. As the deadline approaches with only few hours remaining, we kindly request your feedback on our rebuttal.
>
> Best,
> The authors

---

### Meta-Review · Area_Chair_ksd6 · 2024-12-21

**Metareview:**

In the paper, the authors rigorously examine the factors contributing to posterior collapse in variational autoencoders (VAEs), focusing on the influence of the hyperparameter beta and data size in VAEs.

While there is a consensus among the reviewers that the theories are sound and of interest to ICLR, there are major concerns about the impact and scope of the study, including (1) the model being studied is linear VAE with low latent dimensions. While it is understandable that the simple setting is useful for obtaining useful theoretical insights, it is unclear how the theoretical findings can be generalized to the general settings of VAEs. (2) Several claims in the paper, such as the bad effect of the high value of the hyperparameter beta, had been studied before, which limits the novelty of the current theories in the paper. (3) The experiments are quite poor and minimal, which limits the scope of the theories in the real-world settings.

Given the above major concerns, I recommend rejecting the paper at the current stage. I believe that the paper will be stronger after incorporating the feedback and suggestions of the reviewers.

**Additional Comments On Reviewer Discussion:**

Please refer to the meta-review.

---

### Decision · Program_Chairs · 2025-01-22

Reject